# *Oxime*-functionalized anti-insecticide fabric reduces insecticide exposure through dermal and nasal routes, and prevents insecticide-induced neuromuscular-dysfunction and mortality

Mahendra K. Mohan [1], Ketan Thorat [1], Theja Parassini Puthiyapurayil [1], Omprakash Sunnapu [2], Sandeep Chandrashekharappa [1], Venkatesh Ravula [1], Rajamohammed Khader [1], Aravind Sankaranarayanan [1,3], Hadi Muhammad [1] & Praveen Kumar Vemula [1] ✉

Farmers from South Asian countries spray insecticides without protective gear, which leads to insecticide exposure through dermal and nasal routes. Acetylcholinesterase plays a crucial role in controlling neuromuscular function. Organophosphate and carbamate insecticides inhibit acetylcholinesterase, which leads to severe neuronal/cognitive dysfunction, breathing disorders, loss of endurance, and death. To address this issue, an Oxime-fabric is developed by covalently attaching silyl-pralidoxime to the cellulose of the fabric. The Oxime-fabric, when stitched as a bodysuit and facemask, efficiently deactivates insecticides (organophosphates and carbamates) upon contact, preventing exposure. The Oxime-fabric prevents insecticide-induced neuronal damage, neuro-muscular dysfunction, and loss of endurance. Furthermore, we observe a 100% survival rate in rats when repeatedly exposed to organophosphate-insecticide through the Oxime-fabric, while no survival is seen when organophosphate-insecticide applied directly or through normal fabric. The Oxime-fabric is washable and reusable for at least 50 cycles, providing an affordable solution to prevent insecticide-induced toxicity and lethality among farmers.

Farmers from agrarian countries, specifically South Asian countries, spray insecticides without using protective gear. Acute insecticide poisoning affects more than 300 millions farmers each year, while intentional self-poisoning kills ~150,000 people each year[1–3]. Enzyme acetylcholinesterase (AChE) plays a pivotal role in both central and peripheral nervous system[4,5]. Organophosphate (OP) and carbamate insecticides inhibit the enzyme AChE, resulting in excessive acetylcholine accumulation, which affects muscarinic and nicotinic

[1]Institute for Stem Cell Science and Regenerative Medicine (DBT-inStem), GKVK Post, Bellary Road, Bangalore 560065 Karnataka, India. [2]Sepio Health Private Limited, Bangalore 560065 Karnataka, India. [3]Tata Institute for Genetics and Society (TIGS), inStem, GKVK Post, Bellary Road, Bangalore 560065, India. ✉ e-mail: praveenv@instem.res.in

receptors at synapses. Therefore, inhibition of AChE due to insecticide exposure causes neurological dysfunction, breathing disorder, paralysis, and death in severe cases[6–8]. India is one of the major countries in the world that heavily uses insecticides, and farmers from the Asian-Pacific region are repeatedly exposed to insecticides during farming[8–11]. Electroencephalograms of OP-exposed humans and monkeys revealed that in addition to their acute toxic effects, OPs have also been reported to have long-term effects[12]. Additionally, agricultural workers have reported neuro-behavioral changes even two years after a single exposure to OPs[13,14]. Based on the reported studies, it is known that pre- and postnatal exposures to OPs hinder neurodevelopment in children from agricultural communities, and they are vulnerable to neuropsychiatric disorders[15,16]. Typically used insecticides include OPs (methyl parathion, methamidophos, oxydemeton-methyl, chlorpyrifos, malathion, and primifos-methyl), and carbamates (carbaryl, aldicarb, carbofuran, and ferbam).

While spraying, insecticide exposure primarily occurs through dermal contact and inhalation. Tropical conditions prevent agriculture workers from wearing polythene or non-breathable fabric-based personal protective equipment (PPE), while regular cotton/semi-cotton cloths do not prevent insecticide exposure. Therefore, developing efficient prophylactic technologies to minimize dermal and inhalation exposure to reduce insecticide-induced hazardous effects is a critical unmet clinical need. Currently available headgear, facemasks, gloves, and suits are seldom used, owing to high cost and enormous discomfort under tropical conditions in low and middle income countries[17–19]. Additionally, physical barrier creams were developed to minimize exposure to OPs[20–22]. However, these systems showed limited success as they do not deactivate insecticides, and they physically trap OPs on the surface, which can still enter the system over a longer duration and through hand-to-mouth contact. Hence, physical barrier creams are not adequate to prevent exposure. Therefore, to overcome this limitation, a couple of non-physical barrier dermal systems that can chemically deactivate OPs upon contact were developed by others[23,24] and our group[25]. Due to potential DNA damage caused by metal (CeO$_2$) nanoparticle-based systems in dermal fibroblasts, using these systems as dermal protectants to prevent insecticide exposure is considerably restricted[26,27].

Although earlier we have developed a prophylactic topical gel that can chemically deactivate OPs and limit insecticide-induced toxicity[25], before commercial development of such topical gel for human applications, we wanted to engage with agriculture workers to get their feedback on willingness to adopt this technology. As a community engagement, we interacted with over 200 farmers from 60 rural villages in India, who are suffering from insecticide-induced acute toxicity due to exposure while spraying to see whether they are willing to adopt such prophylactic dermal cream. Although they are thrilled to know about this technology, one critical feedback was received. The need to apply the dermal cream on the entire body every time they spray will be challenging, and compliance might go down over time. This critical feedback inspired us to develop a user-friendly, efficient fabric-based prophylactic technology to prevent exposure while spraying insecticides in the agricultural field. At present, due to comfort, farmers wear regular cotton cloths while farming, which cannot prevent insecticide exposure. Therefore, an ideal solution will be developing an active fabric that can be stitched into a bodysuit and a facemask that can chemically deactivate insecticides upon contact, hence can prevent insecticide exposure through dermal and inhalation routes, respectively (Fig. 1). Additionally, an affordable bodysuit which is wash-resistant, reusable, and low-cost will increase the compliance and have a significant impact on the healthcare of agricultural workers whose purchase power is low.

Herein, we have developed an active fabric by covalently attaching a silyl-oxime-based nucleophile to cellulose that can hydrolyze insecticides upon contact before they enter the body, either in a solution or aerosol form. Earlier, we demonstrated that pralidoxime can hydrolytically deactivate a wide range of OPs and carbamates[25] in addition to their better-known effect of reactivating OP-inhibited acetylcholinesterase[28]. Here, we developed an oxime fabric by covalently connecting silyl-pralidoxime to cellulose fabric. Oxime-fabric, when stitched as a bodysuit and a facemask, can catalytically deactivate a wide range of insecticides (OPs and carbamates), prevent exposure through dermal and inhalation routes, and thereby limit insecticide-induced toxicity and mortality. These results are remarkable, and we aim to develop reusable protective bodysuits and facemasks, which may have an enormous impact on protecting agricultural workers from insecticide-induced toxicity.

## Results
### Preparation of Oxime-fabric and deactivation of insecticides, ex vivo

Commonly used insecticides are from organophosphorus and carbamates classes. The use of insecticides in India is different from that for the different parts of the world. In India OP-based insecticides are 76% of the total pesticides, as compared to 44% globally[29]. On the contrary, herbicides and fungicides are used less compared to insecticides[29,30]. We hypothesized that nucleophile-mediated hydrolysis may deactivate insecticides. Since *N*-hydroxy α-nucleophiles, such as oximes are efficient nucleophiles[31–35], we envisaged that covalent conjugation of pralidoxime based α-nucleophile to the cellulose of fabric would generate an anti-insecticide fabric, Oxime-fabric, which may hydrolyze insecticides upon contact (Fig. 1). Pralidoxime was quaternized with (3-chloropropyl)triethoxysilane, and resulted silyl-oxime was conjugated to cellulose of fabric (Fig. 1 and Methods). A detailed characterization of compounds is given in Supplementary Figs. S1 and S2. To provide further evidence for covalent functionalization of silyl-oxime on the fabric, solid state Cross Polarization Magic Angle Spinning (CP/MAS) $^{13}$C-NMR studies were performed. Pyridinium ring carbons in silyl-pralidoxime compound give peaks between 141–148 δ ppm (Supplementary Fig. S2). Therefore, to find the presence of silyl-pralidoxime upon covalent functionalization on the fabric, $^{13}$C-NMR was recorded for Normal fabric, Oxime-fabric and Oxime-fabric after extensive washing to remove non-covalent bound silyl-pralidoxime. Spectra in Supplementary Figs. S3 and S4 show the presence of silyl-pralidoxime on the fabric. The peaks corresponding to pyridinium group are completely absent in spectra of normal fabric, whereas these peaks were found in spectra of Oxime fabric. Subsequently, Oxime fabric was extensively washed with both detergent and methanol in which silyl-oxime is soluble. However, despite of extensive washing, it did not remove pyridinium peaks, suggesting that silyl-oxime has been covalently attached to the fabric.

Furthermore, we used a bromophenol blue assay to quantify the amount of silyl-pralidoxime attached to the fabric. An acidic dye, Bromophenol Blue (BPB), has been used to quantify the concentration of quaternary ammonium salts on fabric[36,37]. Due to the presence of the pyridinium group, the BPB assay gave the quantitative oxime presence on the fabric (Fig. 2a–c). Post-functionalization of pyridinium-oxime on the fabric, extensive washes were done to remove unfunctionalized oxime. Subsequently, the calculated concentration of pyridinium-oxime on normal fabric and Oxime-fabric was $0.26 \pm 0.14$ and $125 \pm 6\ \mu g/cm^2$, respectively (Fig. 2c), which confirmed the presence of pyridinium-oxime on Oxime-fabric.

The ability of Oxime-fabric to deactivate insecticides and prevent insecticide-induced AChE inhibition was tested using the Franz diffusion apparatus, which is generally used to mimic transdermal penetration (Fig. 2d). OPs are known to inhibit AChE in the blood[38]. An ex vivo assay was performed using rat blood in a Franz diffusion cell to test the ability of Oxime-fabric to reduce MPT-induced AChE inhibition (Fig. 2e). Either normal fabric or Oxime-fabric was placed over a 3.5 kDa dialysis membrane between donor and acceptor

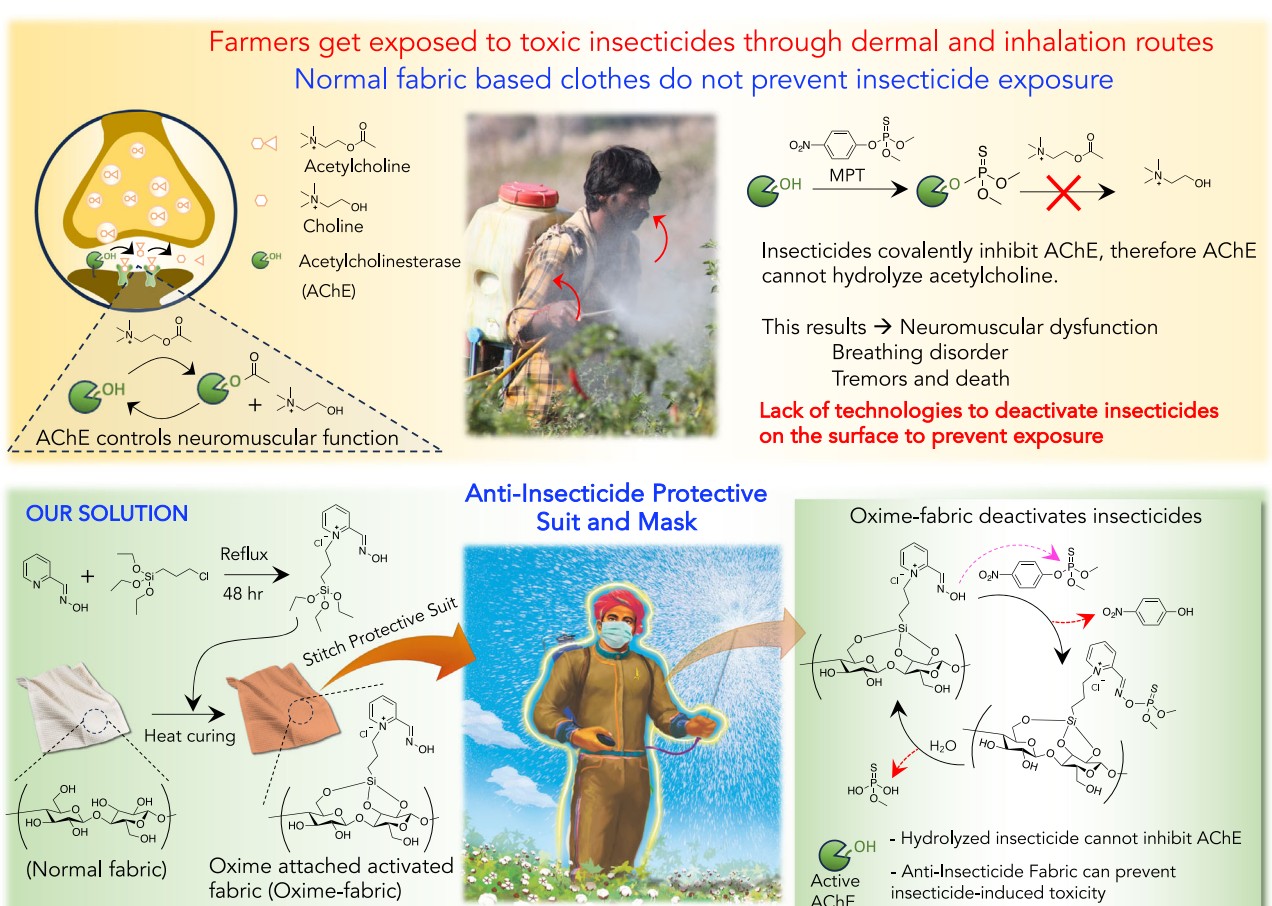

**Fig. 1 | Developing an oxime-fabric that can hydrolyze insecticides to prevent insecticide-induced toxicity.** Top, acetylcholinesterase (AChE) enzyme controls neuromuscular function by hydrolyzing acetylcholine, a neurotransmitter at synapses. While spraying, agriculture workers get exposed to insecticides (methyl parathion, MPT), through dermal and inhalation routes. Upon exposure, organophosphates covalently inhibit AChE, which causes severe toxicity and mortality. bottom, In our approach, α-nucleophile, oxime-based silyl-pralidoxime is covalently attached to the cellulose of fabric to generate Oxime-fabric. Oxime-fabric can be stitched as a bodysuit and a facemask, which, upon contact, can hydrolyze organophosphates into phosphoric acid that cannot inhibit AChE. Therefore, Oxime-fabric could prevent exposure to insecticides, thereby preventing insecticide-induced toxicity and mortality.

chambers. Subsequently, methyl parathion (MPT) was added into the donor chamber, and the acceptor chamber was filled with 1000× diluted rat blood that contains AChE. As one of the controls, only dialysis membrane was used and no fabric was placed between chambers. The chamber's temperature was maintained at 37 °C. The percentage of the AChE activity was quantified using a modified colorimetric Ellman's assay[39] (Methods). In the absence of MPT, no change was observed in the activity of AChE after 3 h (Fig. 2e). The addition of MPT (2.5 µM) in the donor chamber significantly inhibited AChE activity in blood. Direct exposure or the presence of normal fabric did not prevent MPT-induced AChE inhibition (Fig. 2e). On the other hand, Oxime-fabric deactivated MPT before entering into the acceptor chamber, thereby completely reduced MPT-induced inhibition of blood AChE activity.

To test the reusability of Oxime-fabric, after covalent functionalization, Oxime-fabric was subjected to delicate washing cycles mimicking the laundry and washing machine using 0.1% of detergent (Fig. 2f and Methods). After completion of 10, 25, and 50 wash cycles, the concentration of pyridinium-oxime present on the fabric was quantified using BPB assay. Data in Fig. 2g reveal that repeated wash cycles did not reduce the amount of pyridinium-oxime present on the fabric, suggesting that pyridinium-oxime is covalently attached to the fabric and does not leach while washing with the detergent. Subsequently, after different wash cycles (10, 25, and

50), Oxime-fabric was tested for its ability to deactivate MPT to prevent MPT-induced AChE inhibition using Franz diffusion assay. Even after 50 cycles of detergent washes, Oxime-fabric effectively deactivated MPT and prevented AChE inhibition (Fig. 2h). Typically, farmers dry their cloths under the sun. Therefore, to test the stability of Oxime fabric under sunlight, Oxime fabric was exposed to sunlight for 3 days, and subsequently investigated its efficacy to deactivate insecticides. The data in Supplementary Fig. S5 suggest that even after exposed to the sunlight Oxime fabric retained its activity and prevented MPT-induced AChE inhibition.

To test the robustness of Oxime-fabric to deactivate a wide range of insecticides, a similar assay was performed using commercial formulations that contain carbamates and OPs. These formulations are carbaryl (carbamate), Macacid-50 (MPT 50%), Aalphos (Monocrotophos 36%), Raise-505 (Chlorpyrifos 50% + Cypermethrin 5%), and Profex Super (Profenofos 40% + Cypermethrin 4%). In the Franz diffusion assay (Supplementary Fig. S6), Oxime-fabric could hydrolytically cleave all commercial OP formulations to prevent insecticide-induced AChE inhibition. As controls, we have used normal fabric that was not functionalized with silyl-oxime, and commercially available physical barrier Hazmat suits. The data in Supplementary Fig. S6e−h suggest that normal fabric and Hazmat suit fabrics could not prevent insecticide-induced AChE inhibition, while Oxime-fabric could prevent AChE inhibition suggesting that α-nucleophile oxime is essential for

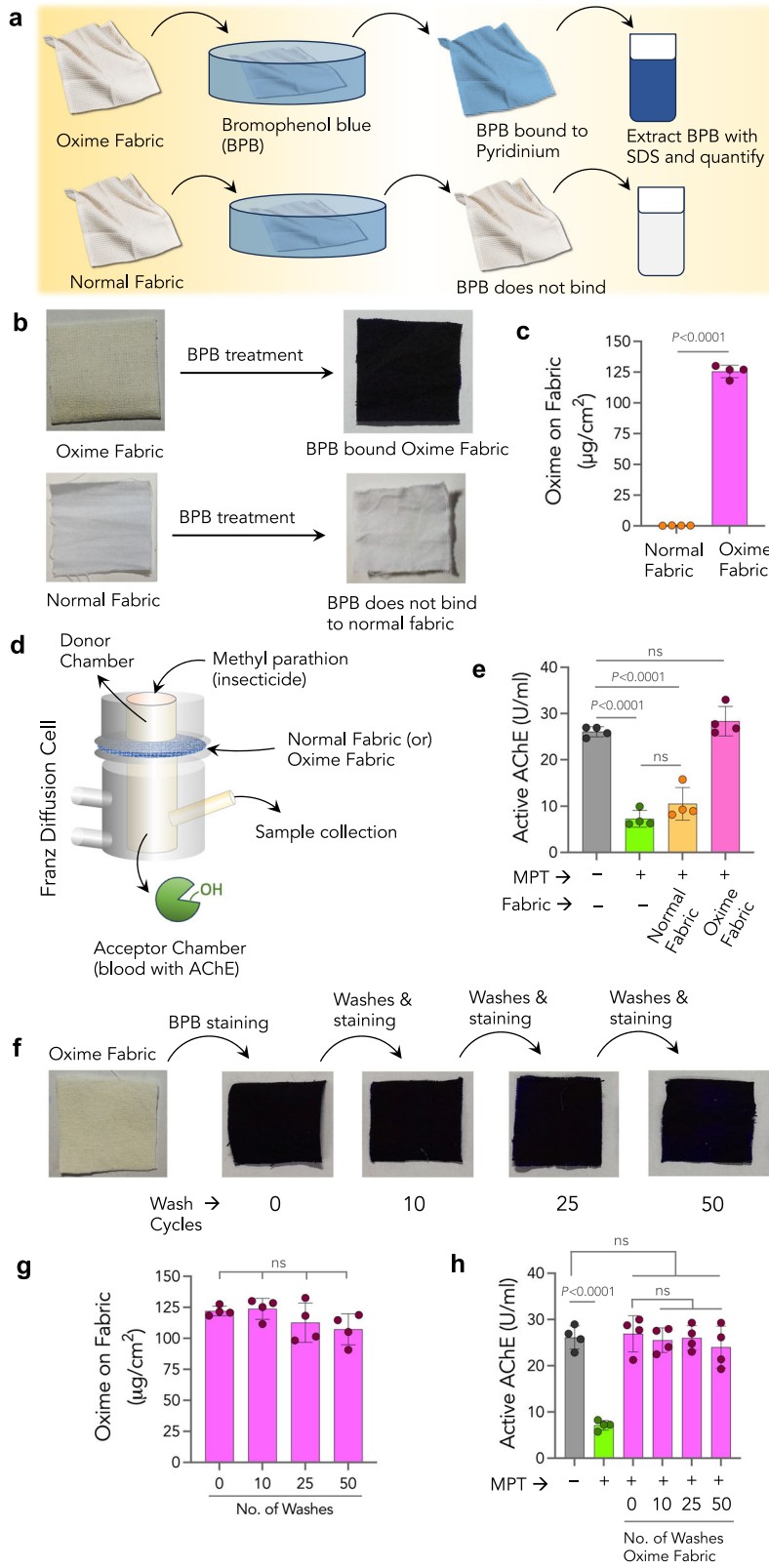

the activity. These results suggest that Oxime-fabric could reduce OP or carbamate insecticide-induced AChE inhibition by chemically deactivating insecticides.

To unambiguously demonstrate that Oxime-fabric does not act like physical barrier cloth, and characterize oxime-mediated hydrolytic cleavage of MPT, normal fabric or Hazmat suit fabric or Oxime-fabric was placed between donor and acceptor chambers that were filled with

phosphate-buffered saline (PBS). Subsequently, MPT was added to the donor chamber, and the concentrations of MPT and its hydrolytically degraded product $p$-nitrophenol were quantified in both chambers at 1 and 180 mins after MPT addition (Methods and Supplementary Fig. S7a, b). In the presence of normal fabric, MPT was quantitatively diffused into the acceptor chamber within 3 h (Supplementary Fig. S7c), whereas Oxime-fabric prevented MPT diffusion

**Fig. 2 | Washable and reusable Oxime-fabric deactivates MPT to prevent MPT-mediated AChE inhibition, ex vivo. a–c** Schematic of bromophenol blue (BPB) assay to quantify the amount of silyl-pralidoxime attached to the fabric (**a**); **b** photographs of Oxime-fabric and normal fabric before and after treating BPB dye, and **c** quantification of silyl-pralidoxime present on fabric, suggested that silyl-pralidoxime (125 μg/cm²) was present on Oxime-fabric, while it was not detectable on normal fabric (*P* < 0.0001; unpaired two-tailed *t* test). **d** Schematic of Franz diffusion cell. **e** The efficacy of Oxime-fabric to prevent methyl parathion (MPT)-induced AChE inhibition, an ex vivo assay was performed using rat blood. Active AChE was measured in unexposed native blood and 3 hr post addition of MPT. The normal fabric could not prevent MPT diffusion into the acceptor chamber, which resulted in significant inhibition of AChE activity (*P* < 0.0001; ordinary one-way ANOVA with Tukey's post hoc analysis). On the contrary, Oxime-fabric could hydrolyze MPT before it diffuses, preventing the MPT-induced inhibition of AChE (*P* = 0.6161; ordinary one-way ANOVA with Tukey's post hoc analysis). **f–h** Oxime-fabric was subjected to delicate washing cycles in a washing machine using a mild non-ionic detergent (0.1%). Photographs of Oxime-fabric after 10, 25, and 50 cycles of washing and stained with BPB dye (**f**), and quantification of silyl-pralidoxime after 10, 25, and 50 cycles of washing (**g**). **h** Franz diffusion assay to investigate the efficacy of multiple cycles of washed Oxime-fabric to hydrolyze MPT to prevent MPT-induced inhibition of AChE activity. Direct exposure to MPT significantly inhibited AChE activity (*P* < 0.0001; ordinary one-way ANOVA with Tukey's post hoc analysis). On the contrary, multiple washing cycles did not cause the leaching of active compound from the fabric and retained its activity. Data are mean ± s.d. (*n* = 4, from independent experime*n*ts). **c** *P* values were determined by two-tailed Student's *t* test with Welch's correction, and for **e, h**, by ordinary one-way ANOVA with Tukey's post hoc analysis, and for **g**, by repeated measures one-way ANOVA by GraphPad PRISM 9, and exact P values are indicated. ns not significant. Source data are provided as a Source Data file.

(Supplementary Fig. S7d). It suggests that normal fabric is not a physical barrier, and MPT can easily pass through it. Interestingly, commercial Hazmat suit fabric, to some extent, prevented the diffusion of MPT, but not quantitatively. Approximately 30% of MPT penetrated and entered into the acceptor chamber (Supplementary Fig. S7e). Data in Supplementary Fig. S7g show that Oxime-fabric was able to hydrolyze MPT into *p*-nitrophenol, which was quantified in both donor and acceptor chambers. Additionally, due to the lack of an Oxime-functional group, either normal fabric (Supplementary Fig. S7f) or Hazmat suit fabric (Supplementary Fig. S7h) could not hydrolyze MPT, therefore, *p*-nitrophenol was not generated in the absence of Oxime-fabric. These results suggest that Oxime-fabric prevents the penetration of MPT, not as a physical barrier but via chemically hydrolyzing MPT.

## Oxime-fabric prevents dermal exposure to insecticides, and reduces insecticide-induced AChE inhibition in blood and tissue, in vivo

To assess the ability of Oxime-fabric to prevent dermal exposure of MPT, and reduce MPT-induced AChE inhibition, 15 Sprague-Dawley rats (10–12 weeks) were randomly divided into three groups (*n* = 5 per group). The dorsal coat was clipped, and was exposed to a single dose of MPT (150 mg/kg) either directly or in the presence of normal fabric (10 × 8 cm²) or through Oxime-fabric (10 × 8 cm²) (Fig. 3a and Methods). MPT is an insecticide categorized as "highly hazardous" (class Ib) by the World Health Organization, as its median lethal dose (LD$_{50}$) falls within 10–100 mg/kg of body weight in rats when exposed via dermal route[40,41]. Therefore, when Sprague-Dawley (SD) rats are exposed to 150 mg/kg of MPT via the dermal route, it is known to cause significant AChE inhibition and toxicity[25]. To investigate the efficiency of Oxime-fabric, the active AChE in the blood at pre-exposure (before) and post-exposure (after three days) was quantified. The data revealed that dermal exposure to MPT either directly or through normal fabric significantly decreased the active AChE in blood. However, in the presence of Oxime-fabric, no such decrease was observed (Fig. 3b). Additionally, after three days, animals were sacrificed, and tissues such as brain, lung, liver, kidney, and heart were isolated, and active AChE was quantified and compared with the active AChE in tissues collected from unexposed naive rats (Fig. 3c–g). Data shown in Fig. 3c–g suggest that dermal exposure of MPT either directly or through normal fabric caused a significant decrease in active AChE levels. However, on the contrary, the same dose of MPT exposure through Oxime-fabric did not decrease the active AChE levels in the studied organs. Therefore, this data corroborates that Oxime-fabric does not act like a physical barrier, but it hydrolytically deactivates OP insecticides, causing prevention of insecticide-induced AChE inhibition, in vivo.

## Oxime-fabric prevents insecticide-induced neuronal damage and perturbed signaling at neuromuscular junctions, in vivo

OP and carbamate insecticides disrupt the functioning of the cholinergic nervous system by inhibiting AChE, which hydrolyses acetylcholine at the synapse. Hence, inhibition of AChE leads to overstimulation of cholinergic receptors, resulting in neuronal excitotoxicity, dysfunction, and disrupted signaling at the neuromuscular junctions. Quantifying the sciatic nerve function is a surrogate method to quantify neuronal damage. Gait analysis is a widely accepted non-invasive method to study sciatic nerve function[42–45]. The analysis of a walking pattern is decided by the posture of the foot, the force exerted, and the angle formed with the ground. These parameters combined will establish a particular print length, toe spread, and intertoe spread of an animal, and these parameters alter significantly when function of the sciatic nerve is impaired[46,47] (Fig. 4a, b). The sciatic functional index (SFI) is an empirically derived formula to evaluate nerve function[44,45] (Methods). Typically, a healthy and unimpaired sciatic nerve function provides SFI values between +11 to −11, whereas SFI values below −20 indicate partial impairment of sciatic nerve function. Gait analysis was performed on 15 animals (SD rats, 10–13 weeks) to evaluate their walking pattern. Before MPT exposure, all animals were trained for 4 days to walk through an alley (Fig. 4a and Methods) and collected the prints of paws (Fig. 4b). Post-training, animals were randomly divided into three groups (*n* = 5 per group), and their SFI was measured. They were subsequently exposed to a single dose of MPT (150 mg/kg) through the dermal route either directly or through normal fabric or Oxime-fabric. Post MPT exposure, the direct exposure and normal fabric groups showed SFI values significantly decreasing to −60 and −36, respectively, which indicate impairment of sciatic nerve function (Fig. 4c). On the contrary, Oxime-fabric protected animals did not show any significant reduction in SFI (+6, Fig. 4c), suggesting complete prevention of impairment in the sciatic nerve function.

OP-insecticide-induced AChE inhibition causes the accumulation of the neurotransmitter acetylcholine at the synapse, leading to perturbed signaling at neuromuscular junctions and overstimulation of muscles. Electromyograms (EMG) were used to monitor if there is signaling impairment at neuromuscular junctions. Upon exposure to MPT (150 mg/kg) through the dermal route either directly or through normal fabric, visible muscular spasms were observed in the animals. For all animals across three groups, we recorded EMG between *spinotrapezius* and *Gluteus maximus* when the animal was awake. EMG of animals exposed to MPT either directly or through normal fabric showed high frequent muscle activity, compared to unexposed animals (Fig. 4d), which suggests the involuntary muscle activity due to accumulation of acetylcholine at synapse. On the contrary, this involuntary muscle activity was completely absent in the animals that were exposed to MPT through Oxime-fabric akin to unexposed animals

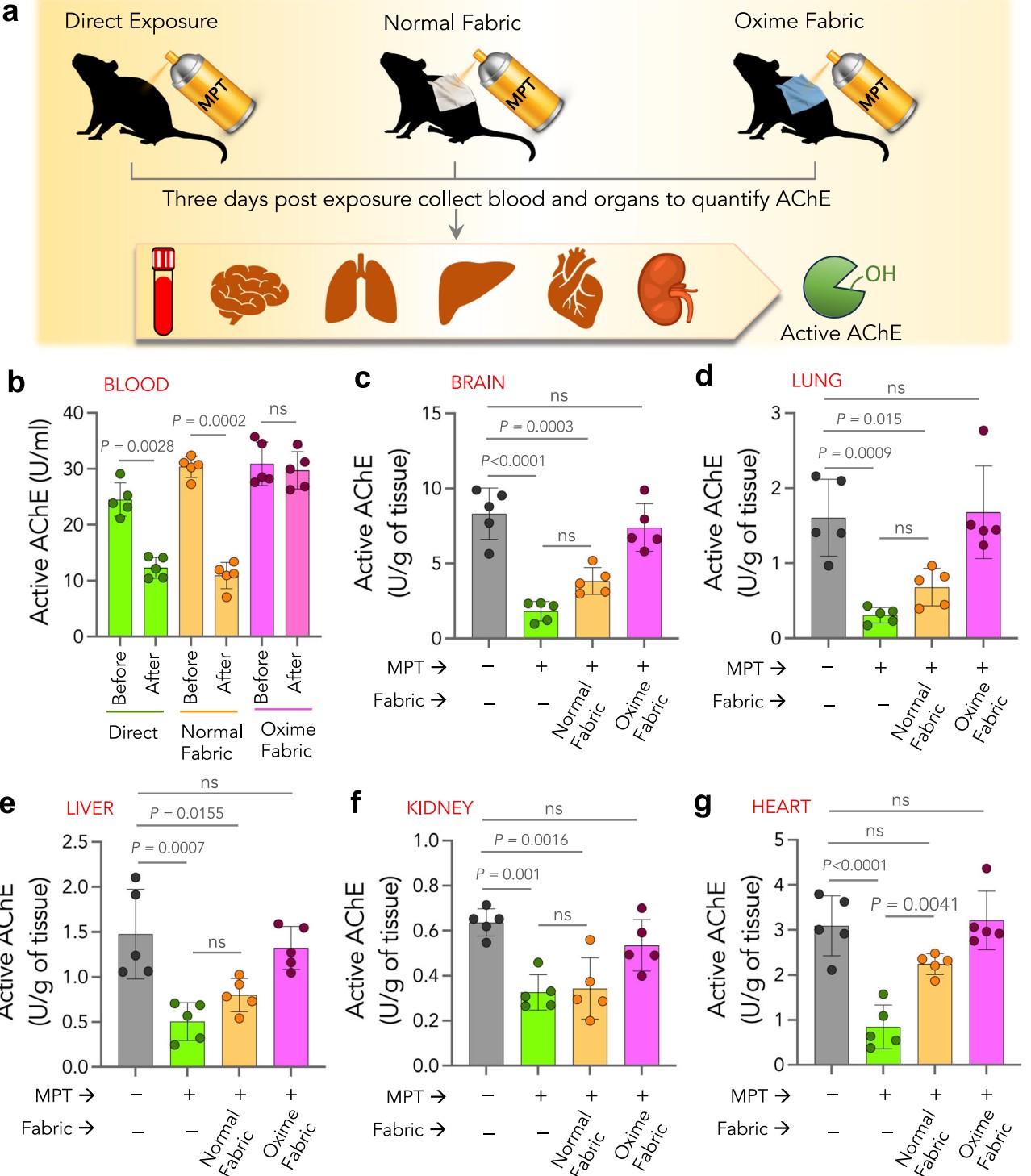

**Fig. 3 | Oxime-fabric prevents acute exposure to a lethal dose of MPT, and reduces AChE inhibition, in vivo. a** Schematic of experimental design: The dorsal coat of Sprague-Dawley (SD) rats was clipped using a hair clipper one day prior to exposure. Organophosphate, methyl parathion (MPT, 150 mg/kg) was applied on the skin either directly or in the presence of normal fabric or Oxime-fabric. **b–g** Before and after 72 h of exposure to MPT, active AChE in the blood (**b**) and internal organs such as brain (**c**), lung (**d**), liver (**e**), kidney (**f**), and heart (**g**) was quantified. Three days post-exposure to MPT, animals were sacrificed, tissues were collected, and the amount of active AChE was quantified using modified Ellman's assay. Dermal exposure of MPT directly or through normal fabric significantly reduced the active AChE in blood and tissues. On the contrary, Oxime-fabric deactivated MPT and prevented MPT-induced inhibition of AChE. Data are mean ± s.d. (*n* = 5 rats per group). **b** *P* values were determined by two-tailed Student's *t* test with Welch's correction, and for **c–g** by ordinary one-way ANOVA with Tukey's post hoc analysis by GraphPad PRISM 9, and exact *P* values are indicated. ns not significant. Source data are provided as a Source Data file.

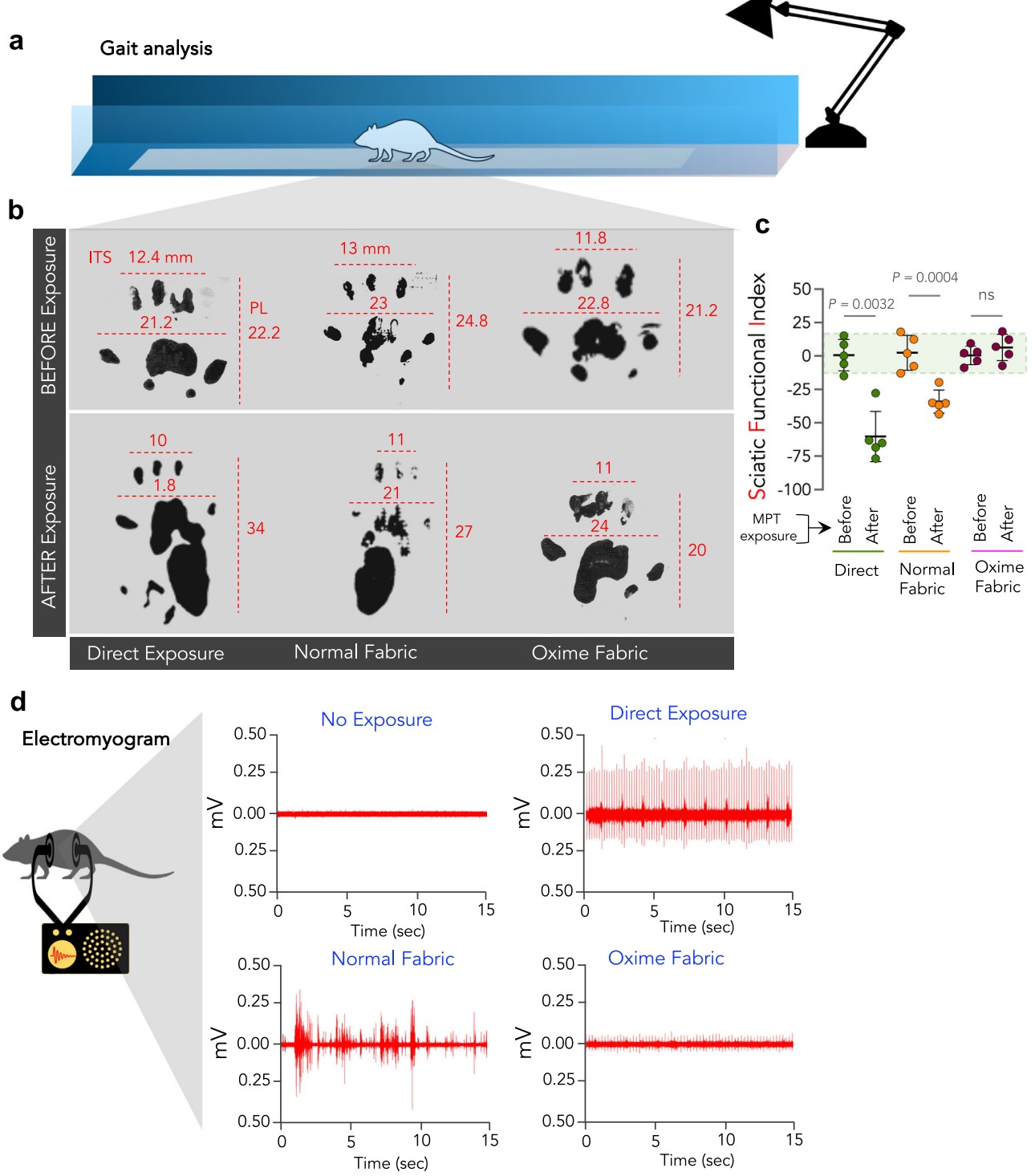

**Fig. 4 | Oxime-fabric prevented sciatic nerve function impairment and involuntary muscle activity, in vivo.** Schematic of an alley in which rats were walked to collect the footprints (**a**). Fifteen Sprague-Dawley rats were randomly divided into three groups (*n* = 5 per group). All paws of the animal were colored with different non-toxic water colors, and trained to walk through an ally (8 cm width, 120 cm length and 10 cm height) leading to its cage. Before exposure to methyl parathion (MPT), footprints of animals in all groups were collected (**b**), and analyzed manually, and the Sciatic function index (SFI) was calculated using the formula reported elsewhere[41,42]. **c** A single dose of MPT (150 mg/kg) was exposed dermally either directly or through normal fabric or Oxime-fabric, and 72 h post-exposure SFI was calculated. Dermal exposure of MPT directly or through normal fabric significantly reduced SFI values, which represents severe impairment of sciatic nerve function, whereas Oxime-fabric prevented such impairment. **d** Electromyograms (EMGs) were recorded before and 72 h post-exposure to MPT for all animals in three groups. Representative graphs are given here. MPT exposure directly and through normal fabric caused overstimulation of neuromuscular signaling and muscle spasms, while Oxime-fabric showed complete prevention of muscle spasms. Data are mean ± s.d. (*n* = 5 rats per group). **c** *P* values were determined by a two-tailed Student's *t* test with Welch's correction by GraphPad PRISM 9, and exact *P* values are indicated. ns not significant. Source data are provided as a Source Data file.

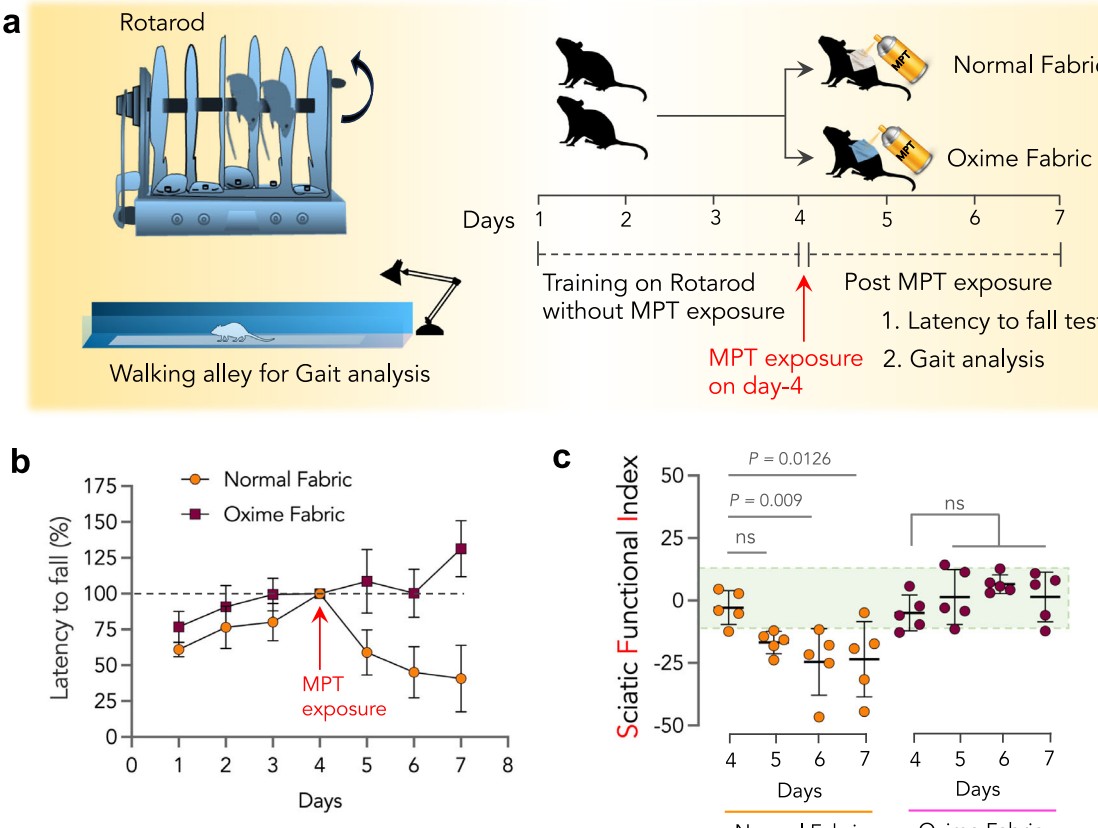

**Fig. 5 | Oxime-fabric prevented insecticide-induced muscular dysfunction and neuronal damage, in vivo. a** Schematic of rotarod and Gait analysis done over time post-exposure to methyl parathion (MPT). Rotarod was used to study muscle function endurance in animals that were exposed to MPT (150 mg/kg) through normal fabric or Oxime fabric. **b** 10 Sprague-Dawley rats were trained on a rotarod for four days, and latency to fall was measured by measuring the time the animal stayed on the rotarod at a constant speed of 20 rpm. The latency to fall obtained on day 4 was considered as 100%, and ten rats were randomly divided into two groups and exposed to MPT through either normal fabric or Oxime-fabric. Post-exposure to MPT, latency to fall has significantly dropped in the normal fabric group, indicating muscle dysfunction and loss of endurance, whereas Oxime-fabric prevented loss of endurance. **c** On day 4, before MPT exposure, the sciatic functional index (SFI) was measured for animals in both groups. Subsequently, after exposing to MPT (150 mg/kg) through either normal fabric or Oxime-fabric, SFI was measured as a function of time. A significant drop in SFI values when exposed to MPT through the normal fabric was observed, indicating neuronal damage, which Oxime-fabric prevented. Data are mean ± s.d. ($n = 5$ rats per group). **c** $P$ values were determined by Repeated Measures one-way ANOVA by GraphPad PRISM 9, and exact $P$ values are indicated. ns = not significant. Source data are provided as a Source Data file.

(Fig. 4d), suggesting that Oxime-fabric could prevent insecticide-induced perturbation in neuronal signaling.

### Oxime-fabric prevents insecticide-induced loss of endurance and motor coordination, in vivo

OP insecticide-induced AChE inhibition impairs signaling at motor end plates, leading to reduced motor coordination and endurance[46-50]. The endurance of rats exposed to MPT was tested using the rotarod experiment. Ten SD rats (10 weeks) were trained on a rotarod for 4 days (Fig. 5a), and during the training, all animals showed similar Lf, latency to fall (Lf = time to fall from rotating rod with 2–20 rpm, Methods). The observed Lf on day 4 was considered as 100%. Simultaneously, during these 4 days, the same animals were trained to walk through an alley to measure SFI, as shown in Fig. 4a, b. The SFI values on day 4 are considered as unexposed controls. On day 4, animals were randomly divided into two groups ($n = 5$ in each group), and exposed to MPT (150 mg/kg) through a dermal route in the presence of either normal fabric or Oxime-fabric (Fig. 5a). Animals that received MPT through normal fabric showed a significant drop in latency to fall from day 5 to 7. At the same time, Oxime-fabric completely prevented such decrease (Fig. 5b). Similarly, post MPT exposure from day 5 to 7, for animals in normal fabric group SFI reduced significantly, while Oxime-fabric prevented a drop in SFI values (Fig. 5c). Cumulatively, these

results suggest that Oxime-fabric could deactivate MPT, and prevents insecticide-induced reduction in endurance and neuronal dysfunction.

### Oxime-fabric prevents repeated insecticide exposure-induced mortality in vivo

As farmers in the field repeatedly get exposed to multiple doses of insecticides, which could cause adverse health effects[51-53], we investigated the robustness of Oxime-fabric to prevent ethyl paraoxon (EPx, an activated OP insecticide)-induced mortality in rats. In survival experiments, to partly mimic the field scenario of multiple exposures, 18 SD rats were randomized into three groups ($n = 6$ per group), and all animals in three groups received EPx through the dermal route (50 mg/kg/day on day 0, and 25 mg/kg/day on days 1–5) once a day until >50% mortality was observed in any two of the groups (Fig. 6a). In case of animals received EPx through either normal fabric or Oxime-fabric, the same fabric was used for all days to test the robust nature of the Oxime-fabric. Animals that received EPx either directly or through normal fabric exhibited the characteristic of OP poisoning symptoms[54], such as muscular fibrillation, salivation, lacrimation, diarrhea, gasping, respiratory distress, and tremors. On the other hand, the animals that received EPx repeatedly through Oxime-fabric did not show signs of toxicity. All animals directly exposed to the EPx group died within 24 h, while all the animals that received EPx through

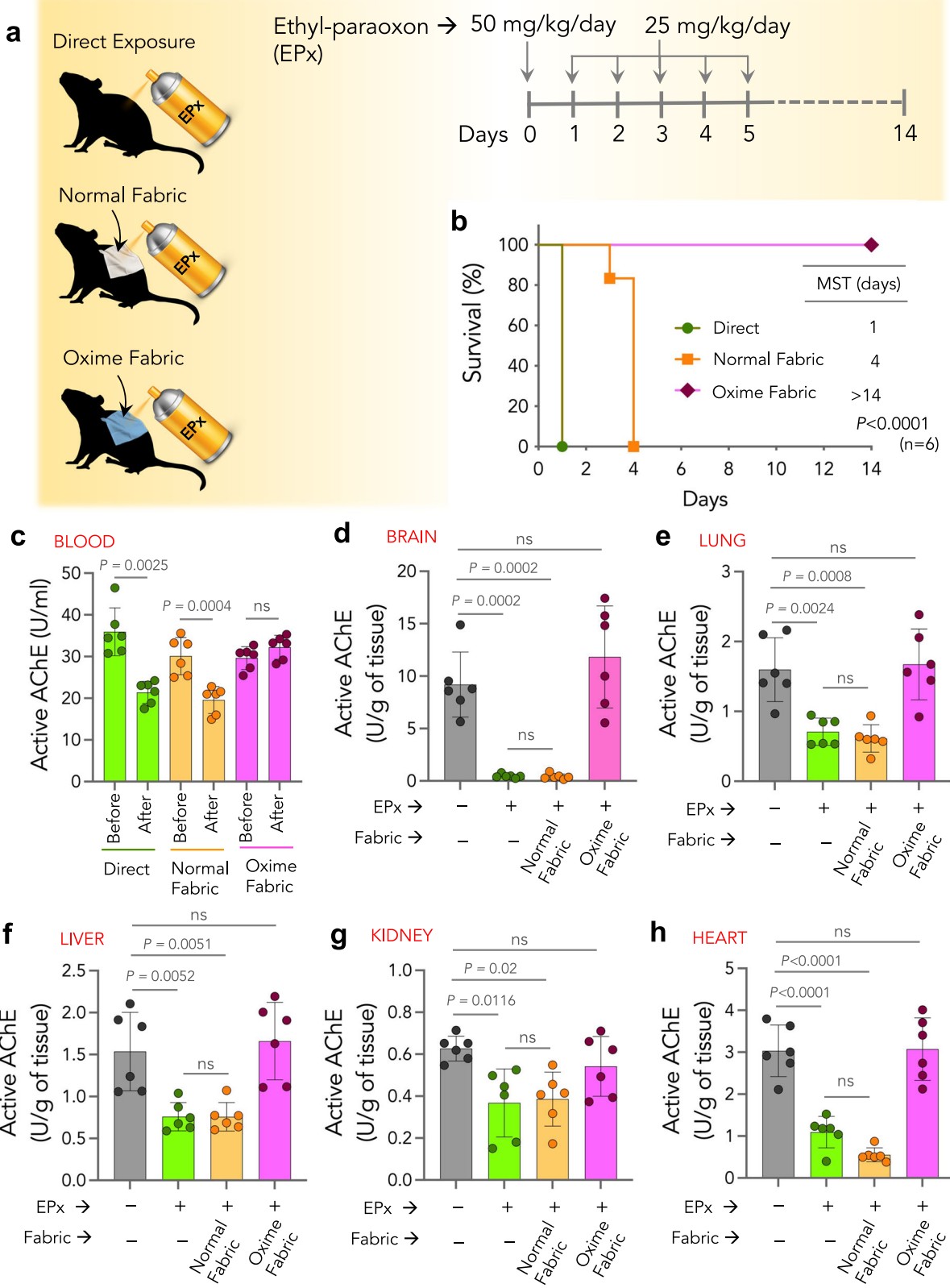

normal fabric died within 4 days, with a median survival time (MST) of 1 and 4 days, respectively. On the contrary, all animals that received EPx through Oxime-fabric survived (Fig. 6b). A remarkable 100% survival rate was observed in the presence of Oxime-fabric ($n = 6$; $P = 0.0001$, Mantel-Cox test). From all the animals, the blood was collected before exposure to EPx (day 0) and at the terminal stage (on day 1 for directly exposed animals, and days 4 and 14 for animals that received EPx

through normal fabric and Oxime-fabric, respectively). Animals that received EPx through Oxime-fabric were tested for any delayed signs of insecticide toxicity for 14 days, and the study was terminated on day 14. Notably, no visible delayed signs of toxicity, such as muscular fibrillation, salivation, lacrimation, diarrhea, gasping, and respiratory distress, were observed. Quantification of active AChE in the blood before and after exposure to EPx revealed that when EPx was exposed

**Fig. 6 | Oxime-fabric prevented mortality during repeated exposure of ethyl-paraoxon (EPx), in vivo. a** Sprague-Dawley rats (10 weeks, males) were randomized in three groups ($n = 6$ rats per group); (i) direct exposure of EPx, (ii) normal fabric + EPx, and (iii) Oxime-fabric + EPx. On day 0, 50 mg/kg/day of EPx and 25 mg/kg/day of EPx were applied dermally for five days. **b** Median survival time (MST) for direct exposure of EPx and EPx received through the normal fabric was 1 and 4 days, respectively, while the Oxime-fabric group did not show mortality ($P < 0.0001$; Mantel-Cox test). **c**–**h** Active AChE in blood and organs was quantified using a modified Ellman's assay. Blood AChE activity dropped significantly in direct EPx and normal fabric groups, while inhibition of AChE was prevented in Oxime-fabric group animals (**c**). Organs were collected either immediately after mortality (for group i and ii animals) or on day 14 (group iii animals), and the amount of active AChE was quantified using Ellman's assay in brain (**d**), lung (**e**), liver (**f**), kidney (**g**), and heart (**h**). Dermal exposure of EPx directly or through normal fabric significantly reduced the active AChE in tissues. On the contrary, Oxime-fabric has deactivated EPx and prevented EPx-induced inhibition of AChE. Data are mean ± s.d. ($n = 6$ rats per group). **b** $P$ values were determined by Mantel-Cox test, and for **c** by two-tailed Student's $t$ test with Welch's correction, and for **d**–**h**, by ordinary one-way ANOVA with Tukey's post hoc analysis by GraphPad PRISM 9, and exact $P$ values are indicated. ns not significant. Source data are provided as a Source Data file.

either directly or through normal fabric, it showed a significant decrease in active AChE (Fig. 6c). At the same time, Oxime-fabric prevented the inhibition of AChE activity (Fig. 6c). Additionally, the tissues were collected from animals that received EPx either directly or through normal fabric immediately after they died. Since 100% survival was observed in the Oxime-fabric group, those animals were sacrificed on day 14, and their tissues were collected. The active AChE in the brain, lung, liver, kidney, and heart was quantified. Data shown in Fig. 6d–h suggest that dermal exposure of EPx either directly or through normal fabric caused a significant decrease in active AChE levels in tissue. However, on the contrary, repeated EPx exposure through Oxime-fabric did not decrease the active AChE levels. It is noteworthy that tissue AChE levels were tested at the terminal point (day 14) for Oxime-fabric group, which might be possible that some of the AChE might have been recovered by day 14. Therefore, these results suggest that Oxime-fabric deactivates OP insecticides in a truly catalytic manner to prevent insecticide-induced toxicity and mortality. Since Oxime-fabric can be washed and reused, as shown in Fig. 2, and it effectively prevents repeated exposure (Fig. 6), we propose that these properties of Oxime-fabric could be beneficial to develop protective suits for farmers who repeatedly get exposed to high doses of insecticides.

## Oxime-fabric prevents insecticide exposure through inhalation route, in vivo

Although dermal exposure accounts for major insecticide toxicity, agriculture workers are also exposed to insecticides through inhalation of air-borne vapors and aerosol/particulate matter. Typically, inhalation exposure nearly accounts for less than 10% of overall exposure[55]. However, inhalation exposure assumes importance, especially for highly volatile insecticides. Therefore, we tested the ability of Oxime-fabric to prevent insecticide exposure through the inhalation route. To test that, we built a chamber where insecticide aerosol can be filled, and restrained animals can inhale aerosols either directly or through the fabric, mimicking the facemask (Fig. 7a and Methods). See the Methods for generating MPT insecticide aerosols. Fifteen SD rats (10-12 weeks) were randomized into three groups ($n = 5$ per group), and they were made to inhale MPT aerosols for 1 h either directly or through normal fabric or Oxime-fabric (Fig. 7a). The blood was collected from animals before exposure and post-exposure 3 days. The quantification of active AChE in the blood at pre and post-inhalation exposure revealed that when animals inhaled MPT aerosols either directly or through normal fabric, active AChE levels in the blood significantly decreased (Fig. 7b). However, no such decrease was observed when animals inhaled MPT aerosols through Oxime-fabric (Fig. 7b). Additionally, three days post-exposure animals were sacrificed, and tissues such as brain, lung, liver, kidney, and heart were isolated and active AChE was quantified and compared with the active AChE in tissues collected from unexposed naïve rats (Fig. 7c–g). The data has shown an interesting trend. In brain tissue, there is no significant difference in active AChE levels in animals that were exposed to aerosol MPT either directly or through normal/Oxime fabrics compared to unexposed animals

(Fig. 7c). On the contrary, the active AChE levels in lung tissue of animals that were exposed to MPT aerosols either directly or through normal fabric significantly decreased compared to unexposed animals, whereas Oxime-fabric prevented such decrease in active AChE levels (Fig. 7d). Active AChE levels in liver, kidney and heart tissue of directly exposed animals were significantly decreased compared to unexposed animals (Fig. 7e–g), however, in the presence of either normal fabric or Oxime-fabric has significantly prevented the reduction in active AChE level in these tissues (Fig. 7e–g). Cumulatively, these results suggest that insecticide exposure through the inhalation route is moderate, and Oxime-fabric can prevent insecticide exposure through the inhalation route.

## Discussion

With the growing demand to increase agriculture productivity to match the food needs of the world's population, the use of OP insecticides has continuously increased over the last six decades[56–58]. Due to low economic strata, agriculture farmers from developing countries manually spray insecticides, whereas, in developing countries, insecticides are usually sprayed using automation. Therefore, the lack of protective gear during spraying is causing direct exposure to insecticides through dermal and nasal routes. OP and carbamate insecticide-induced inhibition of AChE leads to severe acute toxicity, including muscle dysfunction, neuronal dysfunction, respiratory arrest, and cardiac arrest. Additionally, muscle weakness due to acute insecticide exposure contributes significantly to morbidity and consumes healthcare resources such as hospital beds because of prolonged hospitalization[59]. The costs of hospitalization for OP insecticide poisoning have been reported in Sri Lankan and Bangladeshi studies[60–62]. The long-term exposure to insecticides is known to lead several adverse health effects, including risk of metabolic disorders, loss of neuromuscular function[63], neuropsychological performance[64], and neuropsychological dysfunction[13]. The usual protective headgear, facemasks, gloves, and suits are infrequently used due to high cost and discomfort under tropical conditions. Efforts have been made by other groups to develop metal nanoparticle-based active barrier creams[23,24]. However, those systems are not adopted for commercial use due to their potential to cause dermal toxicity. Alternatively, we have developed a *poly*-Oxime-based dermal gel that can efficiently deactivate insecticides to prevent dermal exposure[25]. Although the *poly*-Oxime dermal gel system is efficient, it has two limitations: (i) it cannot prevent exposure through the inhalation route, and (ii) potential non-compliance from agriculture workers as it must be applied all over the body each time they spray insecticides. Additionally, farmers' regular cotton fabric cloths do not prevent insecticide exposure. Earlier attempts to develop protective clothing relied on an adsorptive polyurethane layer impregnated with activated carbon for OPs adsorption[65]. However, such clothes cannot deactivate OPs and must be carefully handled and disposed-off to ensure safety. Cumulatively, the currently available solutions are not adequate to prevent insecticide exposure. Therefore, this critical unmet need encouraged us to develop a nucleophile-attached Oxime-fabric that catalytically hydrolyzes various insecticides upon contact.

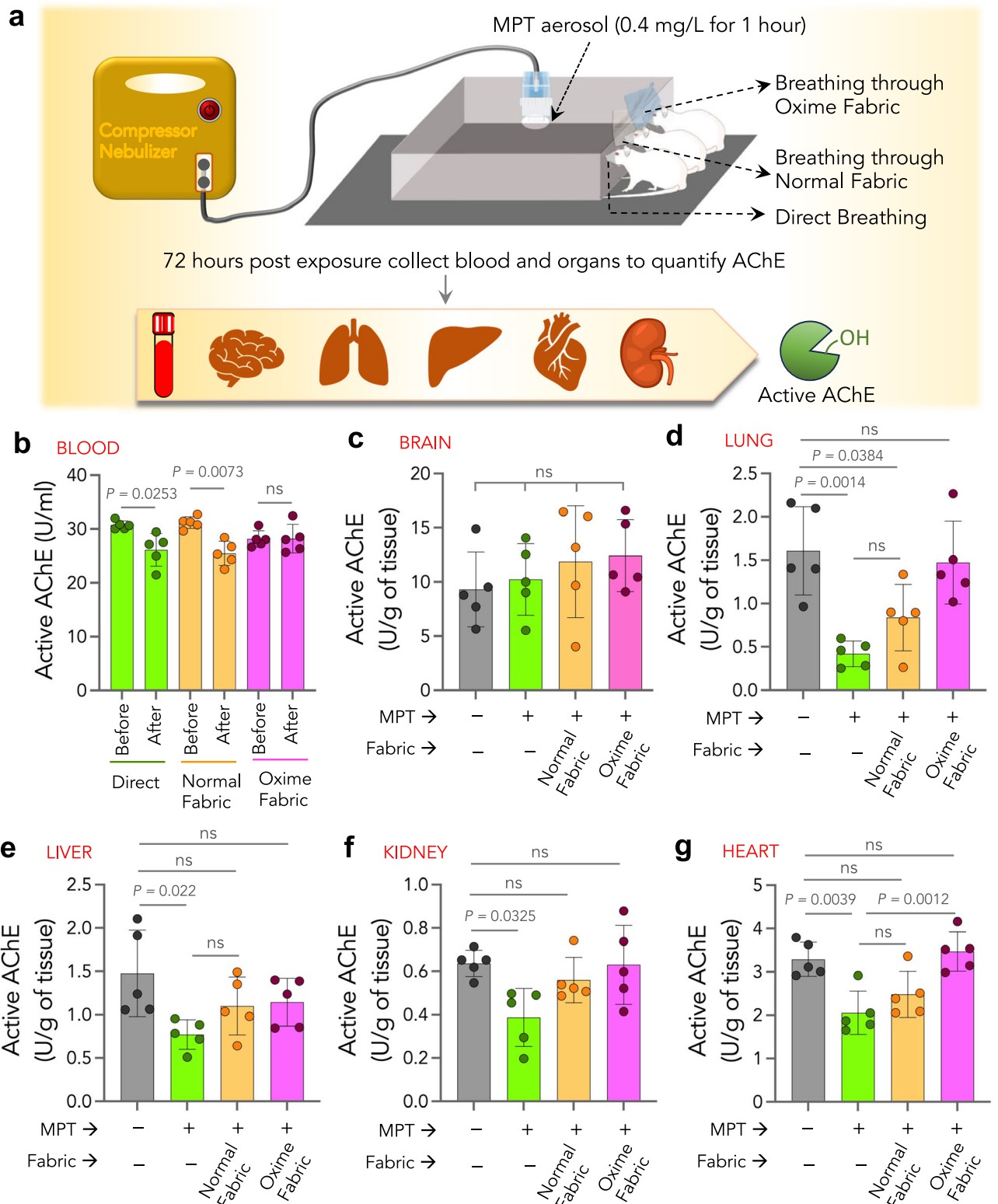

We have optimized a curing process to covalently attach silyl-pralidoxime to the cellulose of fabric to generate Oxime-fabric. The curing process was optimized in a way that the textile industry can adopt the same process using existing machinery to scale up manufacturing of Oxime-fabric in the future, without the need for huge upfront investment to build new infrastructure. Our data suggest that Oxime-fabric can chemically hydrolyze OPs and prevent insecticide-induced AChE inhibition in rat blood, ex vivo. Subsequently, even after 50 rounds of detergent washes, silyl-oxime did not leach from the fabric, and retained its activity to hydrolyze insecticides, suggesting the reusability of Oxime-fabric, which makes Oxime-fabric an economically viable option for agriculture workers. Oxime-fabric can hydrolytically deactivate carbamates and a wide range of commercially available OP formulations, including Macacid-50, Aalphos, Raise-505 and Profex Super. The activity of Oxime-fabric against a wide range of insecticides shows its robustness. Our data suggest that Oxime-fabric, when used as a bodysuit, can hydrolyze OP before it reaches the skin, thus preventing AChE inhibition in blood and internal organs,

**Fig. 7 | Oxime-fabric prevents methyl parathion (MPT) exposure through inhalation route, in vivo. a** Schematic of experimental design: MPT aerosols (0.4 mg/L for 1 h) were generated through a nebulizer that was connected to a closed chamber, where three groups of animals were restrained and inhaled MPT aerosols for 1 h. Fifteen Sprague-Dawley rats were randomly divided into three groups. Group 1 animals inhaled aerosols directly, while group 2 and 3 animals inhaled aerosols through normal and Oxime-fabric, respectively. **b** Active AChE was quantified before and after 72 h of post-exposure, and data showed that MPT aerosol exposure either directly or through normal fabric did not prevent MPT-induced AChE inhibition, while Oxime-fabric prevented inhibition of AChE.

**c**–**g** Three days post-exposure to MPT aerosols, animals were sacrificed, tissues were collected, and the amount of active AChE was quantified in brain (**c**), lung (**d**), liver (**e**), kidney (**f**), and heart (**g**). Nasal exposure of MPT directly or through normal fabric significantly reduced the active AChE in tissues. On the contrary, Oxime-fabric has deactivated the aerosol form of MPT and prevented MPT-induced inhibition of AChE. Data are mean ± s.d. ($n$ = 5 rats per group). **b** $P$ values were determined by two-tailed Student's t-test with Welch's correction, and for **c**–**g** by ordinary one-way ANOVA with Tukey's post hoc analysis by GraphPad PRISM 9, and exact $P$ values are indicated. ns = not significant. Source data are provided as a Source Data file.

including the brain, lung, liver, kidney, and heart. Thus far, there are no examples of fabric-mediated insecticide hydrolysis and prevention of AChE inhibition in vivo. We have done a detailed cost-analysis of Oxime-fabric for commercial use. A comparison analysis between potential price of Oxime-fabric PPE and cost of various commercially available physical barrier PPEs to reduce chemical exposure suggest that Oxime-fabric PPE is cost-effective and affordable (see Supplementary Table S1 and analysis).

Exposure to insecticides causes muscle weakness and loss of endurance[48,59]. As farming demands intense physical activity from agriculture workers, insecticide-induced muscle weakness and loss of endurance can prevent farmers from working at their total capacity, hence, a drop in their productivity. In our preclinical models, a significant loss of endurance and neuromuscular coordination was observed when rats were exposed to insecticides directly or through the normal fabric. On the contrary, Oxime-fabric could deactivate insecticides and quantitatively prevent the loss of endurance, suggesting its potential use to protect agriculture workers from losing their productivity. Furthermore, through Gait analysis, we have demonstrated that Oxime-fabric can prevent insecticide-induced sciatic nerve damage and prevent perturbation of neuromuscular signaling at neuromuscular junctions, hence, can protect neuromuscular junctions. Due to frequent spraying, farmers repeatedly get exposed to insecticides, which causes chronic toxicity and severe adverse health effects[13,63,64]. When rats were repeatedly exposed to ethyl paraoxon directly, or through the normal fabric, all animals died within four days. In contrast, the same doses of repeated exposure through Oxime-fabric could not cause any mortality, and a 100% survival has been observed. Therefore, these results demonstrate the robustness and true catalytic nature of Oxime-fabric to hydrolyze insecticides.

Dermal absorption and respiratory inhalation are the primary routes of exposure to insecticides[66,67]. While applying volatile insecticide products, respiratory exposure occurs. However, for non-volatile insecticides, respiratory inhalation also occurs when insecticides are sprayed in an inhalable aerosol form[55]. Typically, in agriculture farming, approximately 10% of total insecticide exposure occurs through the nasal route, with the rest via dermal absorption[55]. Therefore, we have demonstrated that rats, when inhaling aerosol forms of OPs either directly or through a normal fabric mask, have shown insecticide-induced AChE inhibition. In contrast, the Oxime-fabric mask has prevented the entry of aerosol forms of insecticides and, hence, prevented the inhibition of AChE. The data suggest that Oxime-fabric can prevent insecticide exposure through the respiratory route.

In conclusion, we have developed an Oxime-fabric that hydrolyzes insecticides upon contact and significantly prevents OP and carbamate insecticide-induced AChE inhibition, thereby preventing insecticide exposure-caused muscle dysfunction, neuronal dysfunction, perturbed neuromuscular signaling, and mortality. Oxime-fabric-based bodysuits and facemasks can prevent insecticide exposure through dermal and inhalation routes. Oxime-fabric is highly scalable, washable, reusable, and affordable; therefore, it may increase compliance from agriculture workers and significantly improve their quality of life.

## Methods

All the preclinical experiments presented in this study were performed in compliance with extant regulatory guidelines. All animal studies strictly adhered to institutional and national guidelines for humane animal use. The experimental protocols were approved by the Institutional Animal Ethics Committee (IAEC) at the Institute for Stem Cell Science and Regenerative Medicine (INS-IAS-2020/15(R1)). For all experiments, Sprague-Dawley albino rats at 10 to 14 weeks of age were used. Animals were provided by the animal house at the National Centre for Biological Sciences, Bengaluru. Animals were caged maximum 4 per cage before the experiment and individually after starting the experiment. Food and water were offered *ad libitum*.

### Materials

Bromophenol blue sodium salt (BPB), Pralidoxime, (3-Chloropropyl) triethoxysilane (TCI), methyl parathion, ethyl paraoxon, cypermethrin, Sodium bicarbonate, Ethylene glycol, snake-skin dialysis membrane (MWCO-3500 Da, Thermo Fisher Scientific), Isoflurane (Isotroy), Triton X-100, Sodium dodecyl sulfate (SDS), 5,5′-dithiobis (2-nitrobenzoic acid) (DTNB), Acetylthiocholine iodide (AsChI), Ketamine, and Xylazine were used. Commercial formulations were purchased from local vendors. Macacid-50 (Methyl parathion 50%, Insecticides (India) Ltd., Chopanki, Rajasthan, India, Aalphos (Monocrotophos 36%, Agastya Agro Ltd., Muraharipally, Telangana, India), Raise-505 (Chlorpyrifos 50% + Cypermethrin 5% EC, Agastya Agro Ltd., Muraharipally, Telangana, India), and Profex Super (Profenofos 40% + Cypermethrin 4% EC, Nagarjuna Agrichem Ltd., Punjagutta, Hyderabad, India). Unless mentioned otherwise, all the chemicals were purchased from Sigma-Aldrich.

### Synthesis of Silyl-oxime

A mixture of pralidoxime (16.38 mmol, 2 g, 1eq), (3-chloropropyl)triethoxysilane (16.38 mmol, 3.94 g, 1eq) and KI (8.19 mmol, 1.36 g, 0.5eq) were taken in 50 ml acetone. The reaction mixture was refluxed at 55 °C with stirring for 48 h under an inert nitrogen atmosphere. Once the reaction was complete, the product precipitated upon cooling, and the solid precipitate was separated via filtration and subsequently washed with an excess of ice-cold acetone to eliminate any unreacted starting materials. After drying under vacuum conditions, 0.595 g (yielding 8%) of pure Silyl-Oxime was obtained as a white solid. $^1$*H-NMR* (*DMSO-D$_6$*, 800 MHz): δ: 13.17–13.16 (1H, *s*), 8.99–8.97 (1H, *d*), 8.77–8.76 (1H, *s*), 8.58–8.54 (1H, *t*), 8.45–8.42 (1H, *d*), 8.12–8.09 (1H, *t*), 4.73–4.67 (2H, *m*), 3.46–3.42 (6H, *q*), 1.90–1.84 (2H, *m*), 1.07–1.04 (9H, *t*). $^{13}$*C-NMR* (*D$_2$O, 150 MHz*): δ: 150.45, 149.14, 138.31, 125.19, 122.34, 59.37, 47.63, 25.38, 17.122, 6.67. Mass: m/z: 326.17.

### Covalent functionalization of silyl-oxime on cellulose fabric

A 100 cm$^2$ of cotton fabric weighing 2 grams, with a density of 200 grams per square meter (GSM), was immersed in a 100 ml solution containing 1% sodium bicarbonate. The fabric was then heated to 100 °C for 1 h. Following this treatment, the fabric was thoroughly rinsed with an excess of water and allowed to air dry at room temperature Subsequently, 0.2 gram of silyl-oxime (10%, wt/wt w.r.t. fabric weight)

was dissolved in a 100 ml mixture of water and ethylene glycol (in a 60:40 ratio), and fabric was soaked in this solution for 3 h. After this soaking period, fabric was removed and allowed to air dry for 1 h. Following, it was subjected to a curing process at 70 °C for 30 min followed by an additional curing step at 120 °C for 20 min. Any unconjugated silyl-oxime was removed by washing the fabric with water. The amount of silyl-oxime covalently attached to the fabric was estimated using bromophenol blue assay. Additional characterization has been done using CP/MAS $^{13}$C-NMR studies on a 400 MHz Jeol solid-state spectrometer.

## Bromophenol blue assay

A Bromophenol Blue (BPB) assay was carried with a slightly modified version of known method[36,37]. A fabric sample with a known area of 9 cm$^2$, which had been functionalized, was soaked in 2 ml of BPB solution (10 mM). After soaking for 20 min, fabric was removed and thoroughly washed with regular water to ensure that there was no further leaching of BPB, and it was dried using tissue paper. Subsequently, 1.5 ml of a 1% SDS solution was employed to extract the dye from the fabric, resulting in a solution of dissolved dye. This solution was subjected to UV analysis to measure the absorption at 591 nm, allowing for the determination of the BPB concentration. Using a calibration curve, the concentration of BPB was calculated, which is equivalent to the concentration of silyl-oxime present on the fabric.

## Detergent washing of Oxime-fabric

The Oxime-fabric (silyl-oxime attached fabric) was subjected to a delicate washing cycle which is widely available in commonly used washing machines. In delicate cycle setting, Oxime-fabric was washed for 7 mins with a mild non-ionic detergent (0.1%), which has slow washing and spinning. Between each cycle of washing, the fabric was rinsed with normal cold water and dried for 10 mins. Using this process, Oxime-fabric was washed for 10, 25, and 50 cycles, and subsequently, the amount of silyl-oxime on fabric and its activity to hydrolyze insecticide were tested using Franz diffusion assay after 10, 25, and 50 wash cycles.

## Ex vivo efficacy of Oxime-fabric to prevent insecticide-induced AChE inhibition (Franz diffusion assay)

The dialysis membrane (MWCO 3500 Da) was hydrated overnight in the deionized water at room temperature. Subsequently, it was placed between the donor and acceptor compartments of the Franz diffusion cells (DBK Diffusion apparatus) and clamped to avoid any leakage. The experiment was carried out in three groups (membrane, normal fabric, and Oxime-fabric) comprising four diffusion cells in each set. Between two chambers, 10 cm$^2$ of either normal fabric or Oxime-fabric was placed on the dialysis membrane. Subsequently, the acceptor chamber was filled with 1000× diluted rat blood, and 1 ml of 2.5 μM MPT in phosphate buffer (pH 8) was added into the donor chamber. The temperature of acceptor chamber was maintained at 37 °C. Samples were collected at after 3 h and analyzed using modified Ellman's method[33] (as mentioned elsewhere) for AChE activity as a proxy for MPT exposure. Similarly, prior to MPT exposure, AChE activity was measure, which was considered as unexposed control. In Ellman's assay, we used 5,5'-Dithiobis(2-nitrobenzoic acid)(DTNB), and acetylthiocholine iodide (ASChI) which is specific for AChE. For the colorimetric assay, according to Ellman's method, reaction mixtures were made up of 0.1 mM phosphate buffer (pH 7.4) containing 0.5 mM DTNB and ASChI at a final concentration of 20 mM. The reaction was performed at 25 °C and monitored at 405 nm.

## In vitro efficacy of Oxime-fabric to reduce permeation by hydrolyzing insecticide

Franz diffusion chamber was setup as descried in previous assay. In this assay, acceptor compartment was filled with Phosphate-Buffered Saline (PBS, pH 7.4) instead of rat blood. The experiment was carried out three groups (insecticide exposure through normal fabric or Hazmat suit fabric or Oxime-fabric) comprising four diffusion cells in each set. Between two chambers, 10 cm$^2$ of either normal fabric or Oxime-fabric was placed on the dialysis membrane. 1 μg of MPT was added in the donor chamber. The temperature of acceptor chamber was maintained at 37 ± 0.5 °C using a thermo-static water bath under constant stirring. Samples were withdrawn from acceptor (1 ml) and donor chamber (20 μl) at 1 and 180 mins, and an equal amount of phosphate buffer (pH 7.4) was replaced. The amount of MPT permeated through the membrane, and amount of *para*-nitrophenol formed at each time interval was analyzed by UFLC (Shimadzu, PDA: SPD-M20A, C18 reverse phase column: LC-20AD Prominence Chromatograph). MPT was detected using 60% Acetonitrile in double distilled water (DDW) as mobile phase at 1 ml/min, with a retention time of 5.5 min at 280 nm while maintaining the column at 40 °C. For detection of *para*-nitrophenol, we used 22% acetonitrile, % tri-ethylamine, and 1% tri-fluoro acetic acid in DDW as mobile phase at 0.5 ml/min flow rate through column maintained at 40 °C with a retention time of 3.6 min.

## Protection from a single dose acute dermal exposure to methyl parathion (MPT), in vivo

SD rats (10 weeks) were randomized to three groups ($n = 5$ in each group): (i) direct exposure (MPT 150 mg/kg), (ii) normal fabric (10 × 8 cm, MPT 150 mg/kg), Oxime-fabric (10 × 8 cm, MPT 150 mg/kg). To maximize the contact of the fabric, the dorsal coat was clipped using a hair clipper under mild anesthesia (2.5% Isoflurane) 24 hrs prior to insecticide application taking care not to damage the integrity of the skin. Unless specified otherwise, the total area of 10 cm$^2$ was marked and used for dermal exposure experiments. Before the exposure, 2 μl of blood was collected for the initial active blood AChE quantification. The exposure was performed under mild anesthesia (2.5% isoflurane) for 90 min. After 72 h of post-exposure, animals were sacrificed, and blood and organs such as the brain, lungs, liver, kidney, and heart were collected for AChE quantification. Organ tissue was homogenized (Polytrion PTMR 2100, 1500 rpm) in nine volumes of solution D [1 M NaCl, 1% Triton X-100, 0.01 M Tris-HCl, 0.01 M EDTA (pH 7.4)] and incubated on ice for 1 h, followed by centrifugation at 1888 × g for 45 min. The supernatant was used to quantify the active AChE. The whole blood was diluted 1000× in phosphate buffer for the AChE quantification. The active AChE in blood and organs was quantified by modified Ellman's assay[36]. The active AChE in organs that were collected from the rats that were unexposed to MPT was considered an unexposed control.

## Survival and AChE inhibition study in rats exposed to multiple dermal doses of Ethyl-paraoxon (EPx), in vivo

SD rats (11 weeks) were randomized to three groups ($n = 6$ rats per group): (i) direct exposure (no cloth), (ii) normal fabric (8 × 6 cm), Oxime-fabric (8 × 6 cm), and shaved dorsal hair before the experiment to maximize contact with fabric. In direct exposure group, on day 0, EPx (50 mg/kg/day) was given directly on the skin, and all animals died within 24 h. In normal fabric group, on day 0, 50 mg/kg/day of EPx was given on the top of fabric, and subsequently, from day 1 to 4, 25 mg/kg/day dose of EPx was given. All animals in group died before day 5. In Oxime-fabric group, on day 0, 50 mg/kg/day of EPx was given, and from day 1 to 5, 25 mg/kg/day dose of EPx was given, and animals in this group survived till the termination of the study (day 14). In all animals prior to exposure, 2 μl of blood was collected to quantify initial active blood AChE as an internal control. The insecticide exposure was performed under mild anesthesia (2.5% isoflurane) for 20 min. Immediately after mortality (group 1 and 2) or on the day 14 (group 3), animals were sacrificed, and their organs were collected to quantify AChE level as described above.

## AChE inhibition study in rats exposed to aerosol form of MPT, in vivo

A custom-made polycarbonate and acrylic sheet chamber (40 cm width × 40 cm length × 10 cm height) was designed where the insecticide aerosols can be filled using a nebulizer, and restrained animals can inhale aerosols either directly or through the fabric, mimicking the facemask (Fig. 7a). The chamber was designed to expose 16 animals at once breathing the same aerosol to avoid any experimental differences. SD rats (10 weeks) were randomized into three groups ($n = 5$ rats per group): (i) direct inhalation (no mask) (ii) normal fabric mask, (iii) Oxime-fabric mask and connected to an aerosol chamber with only nose inside the chamber through a 3 cm diameter aperture (with or without fabric mask). MPT (0.4 mg/L) aerosols were generated through flow rate of 6.8 L/min, nebulization rate of 0.3 ml/min using a nebulizer (Model: CNB69011). The exposure was performed under anesthesia (Ketamine 91 mg/kg and Xylazine 9.1 mg/kg were administered intraperitoneally) for 1 h. After 72 h of post-exposure, animals were sacrificed, and their organs and blood were collected to quantify AChE level as described above.

## Gait analysis and electromyogram

Gait analysis was performed on 15 male rats (10 weeks) to evaluate their walking pattern. To get footprints, four paws were colored with different non-toxic watercolors, and the animal was trained to walk through an ally (8 cm width, 120 cm length, and 10 cm height) leading to its home cage. After training, animals were divided randomly into three groups, (i) direct exposure, (ii) normal fabric, and (iii) Oxime-fabric. Footprints were analyzed manually, and Sciatic functional index (SFI) was calculated using following formula[44,45].

$$SFI = \left[ \left( \frac{ETOF - NTOF}{NTOF} \right) + \left( \frac{NPL - EPL}{EPL} \right) + \left( \frac{ETS - NTS}{NTS} \right) + \left( \frac{EIT - NIT}{NIT} \right) \right] \frac{220}{4}$$

Where, $N$ indicates normal/before exposure and E indicates after exposure, TOF: distance to opposite foot; PL: distance from the heel to the third toe, the print length; TS: distance from the first to the fifth toe, the toe spread; IT: distance from the second to the fourth toe, the intermediate toe spread. SFI values were calculated before and 72 h post MPT exposure.

EMG of animals from these three groups and unexposed animals was recorded using a Muscle-Spiker box (Backyard Brains®). Skin surface electrodes were used with adhesive pads and conductive gel to facilitate the recordings. To track the muscle spasms, we recorded EMG between *spinotrapezius* and *Gluteus maximus* when the animal was awake.

## Rotarod test

Total 10 male SD rats (10 weeks) were used to study loss of endurance in MPT-exposed animals and ability of Oxime-fabric to prevent the same. Animals were placed on a Rotarod treadmill (Columbus Instruments; Rotamex-5 1.4, lane width: 9.3 cm, diameter of rod: 7 cm, fall distance: 48.3 cm) and were subjected to a uniform increase in acceleration between 0 to 20 rpm and were allowed to run at 20 rpm as till the animal got tired and fell off the rod, and time of fall was recorded. Rats were trained for 4 days before the exposure where animals were randomly grouped into two groups (i) normal fabric, and (ii) Oxime-fabric. On day 4, in both groups animals were exposed to MPT (150 mg/kg) either through normal fabric or Oxime-fabric. Subsequently, latency to fall was calculated as a time taken to fall on any day with respect to day 4 (before exposure) and was converted to percentage. Simultaneously, Gait analysis was performed on the same 10 animals on day 4 (prior to MPT exposure) and measured SFI as described above. Post MPT exposure, every 24 h SFI was calculated for 72 h.

## Statistical analyses and reproducibility

In experiments with multiple groups, ordinary one-way ANOVA with Tukey's post hoc test was used. In experiments with multiple groups, which are time-course studies, repeated measures of one-way ANOVA were used. The two-tailed Student's $t$ test with Welch's corrections was used to compare two experimental groups. In survival experiments, Mantel–Cox test was used. The probability value ($P$) < 0.05 was considered as a statistically significant difference. Statistical analysis and graphing were performed with GraphPad PRISM 9. Data presented in this manuscript is reproducible, data collected from at least $n = 4$, from three independent experiments.

## Reporting summary

Further information on research design is available in the Nature Portfolio Reporting Summary linked to this article.

## Data availability

All data generated or analyzed during this study are included in this published article and its supplementary information files or are available from the corresponding author upon request. Source Data is provided as a FigShare repository. Source data are provided with this paper.

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

## Acknowledgements

We thank the animal house facility/members and the Central Imaging & Flow Cytometry facility (CIFF) at inStem and NCBS. We thank the NMR facility at NCBS. We thank the Sophisticated Analytical Instrument Facility, a Department of Science and Technology-supported Institute NMR Facility at IISc, for solid-state NMR spectra. We thank Radhika Rao Arasala for helping with setting up in vivo experiments. We thank Mishal Khan for helping with the HPLC study. We thank Mukta Arora, Siraj Chaudhury and Raghavendra Rao from Sepio Health for critical discussions. This project has received funding from the Department of Biotechnology, Govt of India, with a grant no. BT/PR29948/NNT28/1576/2018 to P.K.V. Animal work in the inStem/NCBS Animal Care and Resource Centre was partially supported by the National Mouse Research Resource (NaMoR) grant (grant no. BT/PR5981/MED/31/181/2012;2013-2016;2018) from the Department of Biotechnology.

## Author contributions

P.K.V. conceived and designed the experiments. M.K.M., K.T., T.P.P., O.S., S.C., V.R., R.K., A.S., H.M. designed, performed the experiments, and analyzed the data. M.K.M., T.P.P., S.C., V.R., H.M., R.K. synthesized compounds and functionalized fabric. M.K.M., K.T., O.S., V.R., R.K., A.S. performed in vitro and in vivo experiments. P.K.V. wrote the manuscript, and all authors discussed the results and revised the manuscript. P.K.V. supervised the project.

## Competing interests

P.K.V., K.T., and S.C. hold patents related to this technology: "Composition, materials, and methods for deactivating toxic agents" (Granted Indian patent: 201841006678 and granted Sri Lankan Patent # 20419). All compositions and methods of use described in this manuscript were covered in the patent applications. P.K.V., K.T., and S.C. are inventors of patents that were licensed to Sepio Health, a company that has licensed IP generated by P.K.V., K.T., and S.C., that may benefit financially if the IP is further validated. The interests of P.K.V. were reviewed and are subject to a management plan overseen by his institution in accordance with its conflict of interest policies. P.K.V. and O.S. are equity holders in Sepio Health. The remaining authors declare no competing interests.
