## [Peer Review File · Nature Communications]

Oxime-functionalized anti-insecticide fabric reduces insecticide exposure through dermal and nasal routes, and prevents insecticide-induced neuromuscular-dysfunction and mortalityREVIEWER COMMENTS

Reviewer #1 (Remarks to the Author):

This manuscript describes an innovative approach to protect farmers from dermal and inhalational exposure to OP and carbamate insecticides. It reproduces a series of studies of an oxime-containing gel that were published in 2018 [1]. The gel approach was considered impractical by farmers and therefore the authors developed a cloth with oxime affixed. They show in a series of experiments that it breaks down methyl parathion, to produce the metabolite p-nitrophenol, in a Franz diffusion cell study; and that application of methyl parathion and of ethyl paraoxon through the treated material prevents AChE inhibition, neurotoxicity, and lethality, respectively. This could be a significant preventative approach

Major

1.

I am a little surprised to see oximes used in this way since they are currently used clinically to reactivate OP-inhibited AChE by nucleophilic hydrolysis of the phosphate bound with the serine hydroxyl group, rather than direct hydrolysis of the parent compound. The most definitive review of oximes [4] does not mention this approach at all.

The authors here cite two papers examining the ability of hydroxybenzotriazoles to hydrolyse OP insecticides as the basis for their work. However, oximes are only addressed peripherally and no primary data are produced, especially for the pralidoxime molecule used in this study. The only previous published data I could find on this mechanism of action comes from the authors own oxime-gel work published in 2018.

There should be some discussion in this paper on the fact that this is a completely different use of oximes that has not yet been otherwise addressed in the literature.

2.

The authors claim here and in their 2018 paper that they have shown the oximes to break down both
"a wide range of commercially available organophosphate formulations, including monocrotophos, chlorpyrifos, and carbamates (Carbaryl)." and pyrethroids

Yet the methods state that "all the chemicals were purchased from Sigma-Aldrich" and are not commercial formulations, which would include solvents and other chemicals. In addition, none of the studies in the papers or supplements for this or the 2018 paper list pyrethroids.

The paper actually tests three non-formulated OP active ingredients from Sigma and a single carbamate insecticide. All mention of pyrethrins (or pyrethroids) should be removed from the paper unless there are data to present

This statement also appears to be inaccurate:

"Earlier, we demonstrated that pyridine-2-aldoxime can hydrolytically deactivate a wide range of organophosphates, carbamates, and pyrethrins."

This paper (Thorat 2018) only looked at methyl parathion (and its active metabolite methyl paraoxon). The supplement listing states:

"Fig. S2. Ex vivo Franz diffusion assay with commercial organophosphates"

Yet only chlorpyrifos is otherwise mentioned in the materials section of the paper. It therefore seems that only 2 OP insecticides were examined and no carbamates or pyrethroids

3.

Figure 6 compares the two control (lethal) groups at days 2 to 4 with the intervention (survival) group at day 14 following ethyl parathion exposure. Comparing these organ AChE activities at different days is not an ideal comparison because the OP-inhibited AChE in the intervention group may have recovered by day 14, having been much lower before this (including at 3 days). It is not

a major issue but the different time points should be mentioned as a limitation in the discussion. No earlier data for this OP insecticide are presented.

Minor

1.

"Chemical pesticides are primarily categorized as organophosphate esters, carbamates, and pyrethrins."

Not really. There are many other pesticides, including herbicides and fungicides. The most widely used insecticides worldwide have been the neonicotinoids. Pyrethrins are not common.

The above sentence could say: 'Commonly used insecticides are of organophosphorus, carbamates, and pyrethroid classes'.

2.

"Among them, organophosphates constitute >70% of the total pesticides used in the agricultural fields."

This might be true for India but is not true worldwide. Herbicides are more commonly used than insecticides in many countries. A citation is needed to support this statement if accurate.

3.

"Currently available headgear, facemasks, gloves, and suits are seldom used, owing to high cost and enormous discomfort under tropical conditions in South American and South Asian countries¹⁴⁻¹⁶."

It is not clear why Africa or South East and East Asian countries are excluded from this statement. It relates to all LMICs

4.

The costs of hospitalisation for OP insecticide poisoning have been reported in Sri Lanka and Bangladeshi studies:

Cost to government health-care services of treating acute self-poisonings in a rural district in Sri Lanka.

Wickramasinghe K,
Bull World Health Organ. 2009 Mar;87(3):180-5.

Treatment of self-poisoning at a tertiary-level hospital in Bangladesh: cost to patients and government.

Verma V,
Trop Med Int Health. 2017 Dec;22(12):1551-1560.

Estimating the government health-care costs of treating pesticide poisoned and pesticide self-poisoned patients in Sri Lanka.

Ahrensberg H,
Glob Health Action. 2019;12(1):1692616.

5.

"Gait analysis is a widely accepted non-invasive method to study sciatic nerve function."

Please provide a reference to support this statement

6.

"Pesticides disrupt the functioning of the cholinergic nervous system by irreversibly inhibiting AChE,"

Oximes can reactivate AChE so it is not irreversible

7.

"More than 3 million people worldwide are exposed to organophosphates yearly, accounting for nearly 300,000 deaths annually¹⁻³."

This is inaccurate. Many more people are exposed to OP insecticides. Acute pesticide poisoning probably affects more than 300 millions farmers each year [5] while intentional poisoning kills around 150,000 people each year [6]. It is not clear how many of the cases are due to OP insecticides, but they probably account for about 2/3 of deaths

The references need to be updated.

8.

"Pesticides inhibit AChE, which leads to severe neuronal/cognitive dysfunction, breathing disorders, loss of endurance, and death when exposed to high acute doses."

Only organophosphorus and carbamate insecticides inhibit AChE, not all pesticides.

REFS

1. Thorat K, Pandey S, Chandrashekarappa S, et al. Prevention of pesticide-induced neuronal dysfunction and mortality with nucleophilic poly-Oxime topical gel. *Sci Adv.* 2018;4:eaau1780.
4. Eyer P. The role of oximes in the management of organophosphorus pesticide poisoning. *Toxicol Rev.* 2003;22:165-190.
5. Boedeker W, Watts M, Clausing P, et al. The global distribution of acute unintentional pesticide poisoning: estimations based on a systematic review. *BMC Public Health.* 2020;20:1875.
6. Mew EJ, Padmanathan P, Konradsen F, et al. The global burden of fatal self-poisoning with pesticides 2006-15: Systematic review. *J Affect Disord.* 2017;219:93-104.

Reviewer #2 (Remarks to the Author):

In this manuscript, the authors functionalize cellulose fabric using silyl-oxime to hydrolyze pesticides and prevent pesticide-induced AChE inhibition. The fabric shows high efficiency and high durability. As I am not familiar with the neuromuscular dysfunction, I can not comment on the efficiency and significance of this work for such application. It seems there are reports on using oxime to degrade organophosphates, for example *CS Appl. Nano Mater.* 2023, 6, 5, 3425–3434. Although the chemistry and substrate are different from this study, the idea could be the same by functionalizing the fabric using oxime to degrade organophosphates.

The chemistry used in this study is simple, by silanization on the hydroxy groups of cellulose. However, the author did not show proof of the reaction of oxime onto cellulose. FTIR or solid state NMR could be useful to prove the covalent linkage of the oxime with cellulose.

Overall it is an interesting work and could be considered to publish in *Nature Communication* after some revision. I suggest the authors to resort to experts in the neuromuscular dysfunction for the novelty of this work.

Reviewer #3 (Remarks to the Author):

Oxime-functionalized anti-pesticide fabric reduces pesticide exposure through dermal and nasal routes, and prevents pesticide-induced neuromuscular- dysfunction and mortality

Dear authors,

General questions:

The article aims to demonstrate the catalytic capacity of a tissue attached to silyloxime that prevents acetylcholinesterase inhibition by organophosphates and carbamates. The article is well-written in all sections. However, there are some questions addressed to the authors that require changes or further clarification.

Abstract:

Lines 23-26: "Oxime-fabric, when stitched as a bodysuit and facemask, upon contact efficiently deactivates pesticides (organophosphates, carbamates, and pyrethrins), and prevents exposure through skin and nasal routes, respectively". How can oxime-fabric catalytically deactivate pyrethrins? Please, clarify it.

Introduction:

Lines 42-45: In the 1st paragraph: "Therefore, inhibition of AChE due to pesticide exposure causes neurological dysfunction, breathing disorder, paralysis, and death in severe cases from the Asian-Pacific region are repeatedly exposed to pesticides during farming", the authors use pesticide as insecticide synonym. Please, correct it.

Lines 88-102: The introduction should address the specific research problem, the gaps, specify objectives and what is intended to be done. However, the last paragraph of the introduction describes and discusses the results and presents conclusions. Please, move the last paragraph to suitable sections (results and discussion).

Lines 43-49: In "Therefore, inhibition of AChE due to pesticide exposure causes neurological dysfunction, breathing disorder, paralysis, and death in severe cases... Electroencephalograms of OP-exposed humans and monkeys revealed that in addition to their acute toxicity effects, OPs have also been reported to have long-term effects. Additionally, agricultural workers have reported neuro-behavioural changes even two years after a single exposure to OP", the author included an old reference on the sarin chemical weapon toxic effect in SNC. However, there are recent literature on organophosphate insecticide long-term exposure and neurological disorders. See in: <https://doi.org/10.1016/j.tox.2022.153407>

Lines 51-54: About "Typically used pesticides/insecticides include organophosphates (methyl parathion, ethyl paraoxon, methamidophos, oxydemeton-methyl, chlorpyrifos, malathion, and primifos-methyl), carbamates (carbaryl, aldicarb, carbofuran, and ferbam), and pyrethrins/pyrethroids". Please, remove pesticide in the sentence.

Results:

Lines 113-114: "Chemical pesticides are primarily categorized as organophosphate esters, carbamates, and pyrethrins." Chemical pesticide classes include types of insecticides, fungicides, herbicides so on. Please, in the sentence above change "pesticide" to "insecticides".

Lines 113-118: About "Chemical pesticides are primarily categorized as organophosphate esters, carbamates, and pyrethrins. Among them, organophosphates constitute >70% of the total pesticides used in agricultural fields. Thus, developing a robust nucleophile that can hydrolyze organophosphate esters, carbamates, and pyrethrin esters will enable the deactivation of pesticides. A wide range of a-nucleophiles are known to hydrolyze organophosphates.": The results section summarizes and presents the study findings to put them in context with research question. The study's data should be presented in a logical sequence without bias or interpretation. Please, consider removing them and including them in the discussion section.

Still on lines 113-114: "...organophosphates constitute >70% of the total pesticides used in the agricultural fields.". Please, include the reference.

Lines 147-149: About: "To test the robustness of Oxime-fabric to deactivate a wide range of pesticides, a similar assay was performed using carbamates and a wide range of commercial organophosphate formulations that contain organophosphates and pyrethrins as well.": In the supplementary material, I did not find any results on the catalysis of insecticide formulation containing pyrethroids by oxime fabric.

Lines 126-127: About: "Due to the presence of the pyridinium group, the BPB assay gave the

quantitative oxime presence on the fabric (Fig. 2a, b)". The results in figure 2b show the presence or not of color in the fabric. Consider only including Figure 2a as indicative of quantitative results.

Lines 181-192: The discussion of Figure 2 f and h is presented elsewhere after the presentation of complementary Figure 5. Please, include the results in the correct order.

Lines 147-162: The results description of supplementary tables 3 and 4 is very wordy. I think the author should be direct.

Lines:118-221: The results section needs to be straightforward. In "Pesticides disrupt the functioning of the cholinergic nervous system by irreversibly inhibiting AChE, which hydrolyses acetylcholine at the synapse. Hence, inhibition of AChE leads to overstimulation of cholinergic receptors, resulting in neuronal excitotoxicity, dysfunction, and disrupted signaling at the neuromuscular junctions.": Please. Consider removing the sentence from the results section and including it in the discussion.

Figure 4 b is not mentioned in the results section (photos of paws)

Lines 270-273: About "Oxime-fabric prevents repeated pesticide exposure-induced mortality in vivo. As farmers in the field repeatedly get exposed to multiple doses of pesticides, which could be lethal and can cause mortality, we investigated the robustness of Oxime-fabric to pesticide prevent, ethyl paraoxon (EPx)-induced mortality in rats". Please include references that affirm occupational exposure to multiple pesticide doses could be lethal and can cause mortality.

Lines 240-242: About "Pesticide-induced AChE inhibition causes the accumulation of the neurotransmitter acetylcholine at the synapse, leading to disturbed signaling at neuromuscular junctions and overstimulation of muscles". The author uses pesticide as a synonym for insecticide. Please, replace pesticide with insecticide.

Discussion:

General questions:

- In tropical developing countries, people often place their clothes in the sun to dry. The authors did not evaluate the oxime fabric's thermal stability and photodegradation. The authors should discuss these issues in the limitations section.

- It is worth mentioning some diseases associated with long-term insecticide exposure in the discussion section.

- P-nitrophenol is the catalytic product of organophosphates. This substance is absorbed through the skin (www.epa.gov/sites/default/files/2016-09/documents/4-nitrophenol.pdf; <https://www.atsdr.cdc.gov/toxprofiles/tp50.pdf>). Would p-nitrophenol be toxic to rats and humans? It is important to discuss this issue as a possible study limitation or future issue to investigate.

Lines 393-395: Please include references to "Due to frequent spraying, farmers repeatedly get exposed to pesticides, which causes chronic toxicity and mortality".

We thank all the reviewers and the editor for their positive responses and for finding our work innovative, novel, and of potential interest. The reviewers provide a critical feedback to improve the quality of manuscript. Thus, we have revised the manuscript and performed the suggested experiments to answer the reviewers' comments. Please see below for our response to each point that the reviewers have raised.

Reviewer # 1. We thank the reviewer for critically reading our manuscript and for appreciating that the current work is innovative approach to protect farmers from dermal and inhalational exposure to OP and carbamate insecticides. We thank her/him for providing constructive feedback. Please see below for our response to each point that has been raised by the referee.

Major comments:

Comment R1-1: I am a little surprised to see oximes used in this way since they are currently used clinically to reactivate OP-inhibited AChE by nucleophilic hydrolysis of the phosphate bound with the serine hydroxyl group, rather than direct hydrolysis of the parent compound. The most definitive review of oximes[4] does not mention this approach at all.

The authors here cite two papers examining the ability of hydroxybenzotriazoles to hydrolyse OP insecticides as the basis for their work. However, oximes are only addressed peripherally and no primary data are produced, especially for the pralidoxime molecule used in this study. The only previous published data I could find on this mechanism of action comes from the authors own oxime-gel work published in 2018.

There should be some discussion in this paper on the fact that this is a completely different use of oximes that has not yet been otherwise addressed in the literature.

Reply: We thank the reviewer for this critical comment. Oximes belong to the family of alpha-nucleophiles such as N-hydroxy compounds. Hence they have the ability to rapidly attack phosphate esters. In the literature, they are typically used to reactivate AChE by nucleophilic hydrolysis of the phosphate-bound serine. We have leveraged its ability to attack phosphate ester to hydrolyse OPs and carbamates to deactivate them, therefore, it could be used to prevent exposure as a preventive measure. Earlier we have systematically demonstrated that N-hydroxy group in Oxime is key for hydrolysing OPs (*Sci Adv.* 2018;4:eaau1780). In that study, we did demonstrate that when N-hydroxy converted into N-methoxy, it completely loses its ability to hydrolyse OPs, which clearly demonstrate that N-O⁻ is the nucleophile to attack phosphate esters to hydrolyse them. Additionally, recently in one of the publication Biswas et al (*ACS App. Nano Mater.* 2023;6:3425-3434) have shown that oxime-attached to a polymer could hydrolyse organophosphates akin to our previous demonstration with oxime-gel. Taken together, these examples suggesting that N-hydroxy in Oxime is key to hydrolyse insecticides.

Comment R1-2: The authors claim here and in their 2018 paper that they have shown the oximes to break down both “a wide range of commercially available organophosphate

formulations, including monocrotophos, chlorpyrifos, and carbamates (Carbaryl)” and pyrethroids.

Yet the methods state that “all the chemicals were purchased from Sigma-Aldrich” and are not commercial formulations, which would include solvents and other chemicals. In addition, none of the studies in the papers or supplements for this or the 2018 paper list pyrethroids.

The paper actually tests three non-formulated OP active ingredients from Sigma and a single carbamate insecticide. All mention of pyrethrins (or pyrethroids) should be removed from the paper unless there are data to present.

This statement also appears to be inaccurate: “Earlier, we demonstrated that pyridine-2-aldoxime can hydrolytically deactivate a wide range of organophosphates, carbamates, and pyrethrins.”

This paper (Thorat 2018) only looked at methyl parathion (and its active metabolite methyl paraoxon). The supplement listing states:

“Fig. S2. Ex vivo Franz diffusion assay with commercial organophosphates”
Yet only chlorpyrifos is otherwise mentioned in the materials section of the paper. It therefore seems that only 2 OP insecticides were examined and no carbamates or pyrethroids.

Reply: We thank the reviewer for pointing out this critical point. We apologize for the oversight while writing this part in Methods section.

In fact, the commercial formulations were purchased from the pesticide vendor shops in rural areas. We completely missed giving the commercial formulation details in the Methods. Instead of giving the names and composition of commercial formulation, we gave only active insecticide names. Therefore, we have corrected that mistake and revised the Methods. Now we have included the details of commercial formulations. The details are following.

We have tested four commercial formulations; Macacid-50, Aalphos, Raise-505, and Profex Super. These formulations contain various organophosphates, and mixture of organophosphates, pyrethrins and emulsifiers. **Macacid-50** (Methyl parathion 50%, Insecticides (India) Ltd., Chopanki, Rajasthan, India), **Aalphos** (Monocrotophos 36%, Agastya Agro Ltd., Muraharipally, Telangana, India), **Raise-505** (Chlorpyrifos 50% + Cypermethrin 5% EC, Agastya Agro Ltd., Muraharipally, Telangana, India), and **Profex Super** (Profenofos 40% + Cypermethrin 4% EC, Nagarjuna Agrichem Ltd., Punjagutta, Hyderabad, India).

The data included in the revised supplementary information (**Supplementary Fig. S6**). Same data is given here in **Figure R1**. Accordingly, we have revised the text in the revised manuscript.

Figure R1. Oxime-fabric deactivates commercial pesticide formulations, and prevents AChE inhibition, *ex vivo*. **a**, Photograph image of commercial formulations used in this study. **b**, The efficacy of Oxime-fabric to prevent commercial insecticide formulations-induced AChE inhibition, an *ex vivo* assay was performed using rat blood. AChE containing rat blood was placed in the acceptor chamber. In the donor chamber, various insecticides such as carbamate (c), MPT (d), Macacid-50 (e), Aalphos (f), Raise-505 (g), and Profex Super (h) were added in the presence of normal fabric or Hazmat suit fabric or Oxime-fabric. Active AChE was

measured in unexposed native blood and 3 hr post addition of pesticide formulations in these groups. The normal fabric and Hazmat suit fabric could not prevent diffusion of commercial pesticide formulations into the acceptor chamber, which resulted in significant inhibition of AChE activity. On the contrary, Oxime-fabric could hydrolyze carbamate and insecticides in commercial formulations before they diffuse, hence prevented the insecticide-induced inhibition of AChE. Data are mean \pm s.d. ($n = 4$, from independent experiments). For **c-h**, P values were determined by ordinary one-way ANOVA with Tukey's post hoc analysis by GraphPad PRISM 9, and exact P values are indicated. ns = not significant. Source data are provided as a Source Data file.

Comment R1-3: Figure 6 compares the two control (lethal) groups at days 2 to 4 with the intervention (survival) group at day 14 following ethyl parathion exposure. Comparing these organ AChE activities at different days is not an ideal comparison because the OP-inhibited AChE in the intervention group may have recovered by day 14, having been much lower before this (including at 3 days). It is not a major issue but the different time points should be mentioned as a limitation in the discussion. No earlier data for this OP insecticide are presented.

Reply: We thank the reviewer for this critical suggestion. We do agree that tissue AChE levels for the two controls (direct OP exposure or through Normal Fabric) groups were at days 2 and 4, respectively, whereas interventional (Oxime Fabric) group is at day 14.

We do agree that it is a possibility that OP-inhibited AChE in the intervention group may have recovered by day 14 for tissue levels. In the revised manuscript, we have mentioned as a limitation in the results section.

However, in a single dose exposure of OP experiment (**Figure 3**), tissue AChE levels are compared on the same time-point, day-3. The data clearly suggests that on day-3 tissue AChE levels are higher in intervention group (Oxime-fabric) and comparable with unexposed control group, whereas on same timepoint (day-3) tissue AChE levels are significantly lower in two control groups (direct exposure and normal fabric).

Accordingly, we have modified the revised manuscript.

Minor Comments:

Comment R1-4: "Chemical pesticides are primarily categorized as organophosphate esters, carbamates, and pyrethrins." Not really. There are many other pesticides, including herbicides and fungicides. The most widely used insecticides worldwide have been the neonicotinoids. Pyrethrins are not common. The above sentence could say: 'Commonly used insecticides are of organophosphorus, carbamates, and pyrethroid classes'.

Reply: We thank the reviewer for this point. As reviewer suggested, we have modified the sentence accordingly.

Comment R1-5: "Among them, organophosphates constitute >70% of the total pesticides used in the agricultural fields." This might be true for India but is not true worldwide. Herbicides

are more commonly used than insecticides in many countries. A citation is needed to support this statement if accurate.

Reply: We thank the reviewer for this point. Based on the reviewer suggestion, we have modified the statement as following.

“The use of pesticides in India is different from that for the different parts of the world. In India organophosphate based insecticides are 76% of the total pesticides, as compared to 44% globally. On the contrary, herbicides and fungicides are used less compared to insecticides. (Aktar et al *Interdisc Toxicol.* 2009;2(1):1–12; Agnihotri, Nameeta. “Pesticide Consumption in Agriculture in India - an Update.” *Pesticide Research Journal* 2000;12(1):150-155.

In the revised manuscript, new sentence and references have been added.

Comment R1-6: “Currently available headgear, facemasks, gloves, and suits are seldom used, owing to high cost and enormous discomfort under tropical conditions in South American and South Asian countries14-16.” It is not clear why Africa or South East and East Asian countries are excluded from this statement. It relates to all LMICs

Reply: We thank the reviewer for this point. As reviewer suggested, we have modified the sentence accordingly.

It has been modified as “Currently available headgear, facemasks, gloves, and suits are seldom used, owing to high cost and enormous discomfort under tropical conditions in low and middle income countries.”

Comment R1-7: The costs of hospitalisation for OP insecticide poisoning have been reported in Sri Lanka and Bangladeshi studies:

- Cost to government health-care services of treating acute self-poisonings in a rural district in Sri Lanka. (Wickramasinghe K, *Bull World Health Organ.* 2009 Mar;87(3):180-5.)

- Treatment of self-poisoning at a tertiary-level hospital in Bangladesh: cost to patients and government. (Verma V, *Trop Med Int Health.* 2017 Dec;22(12):1551-1560)

- Estimating the government health-care costs of treating pesticide poisoned and pesticide self-poisoned patients in Sri Lanka. (Ahrensberg H, *Glob Health Action.* 2019;12(1):1692616.)

Reply: We thank the reviewer for providing these references. In the revised manuscript, we have included these references with appropriate statement.

Comment R1-8: “Gait analysis is a widely accepted non-invasive method to study sciatic nerve function.” Please provide a reference to support this statement

Reply: As reviewer suggested, we have added following references.

“Gait-stance duration as a measure of injury and recovery in the rat sciatic nerve model.” *Journal of Neuroscience Methods* 1994;52:47-52.

“A new approach to assess function after sciatic nerve lesion in the mouse—Adaptation of the sciatic static index.” *Journal of Neuroscience Methods* 2007;161:259-264.

“Walking track analysis: An assessment method for functional recovery after sciatic nerve injury in the rat.” *Folia Morphol.* 2009;68:1–7.

“An index of the functional condition of rat sciatic nerve based on measurements made from walking tracks.” *Exp. Neurol.* 1982;77:634–643.

Comment R1-9: “Pesticides disrupt the functioning of the cholinergic nervous system by irreversibly inhibiting AChE,”. Oximes can reactivate AChE so it is not irreversible.

Reply: We thank the reviewer for mentioning this point. Although, if oxime treatment can be provided sufficiently soon after OP exposure, it can reactivate AChE. However, the AChEs inhibited with the organophosphate insecticides are known to undergo the phenomenon of ‘aging’, i.e., the irreversible inhibition of the enzyme. The phosphorylated cholinesterase enzyme undergoes dealkylation of the alkoxy group leading to irreversible binding with the OPs, and therefore making the enzyme irresponsive to reactivation by oxime-based therapeutics.

- Sun M, Chang Z, Shau M, Huang R, Chou T. “The mechanism of ageing of phosphorylated acetylcholinesterase.” *Eur J Biochem.* 1979;100(2):527-30.

- Curtil C, Masson P. Le vieillissement des cholinestérases après inhibition par les organophosphorés [Aging of cholinesterase after inhibition by organophosphates]. *Ann Pharm Fr.* 1993;51(2):63-77. French. PMID: 8250487.

Comment R1-10: “More than 3 million people worldwide are exposed to organophosphates yearly, accounting for nearly 300,000 deaths annually¹⁻³.”

This is inaccurate. Many more people are exposed to OP insecticides. Acute pesticide poisoning probably affects more than 300 millions farmers each year [5] while intentional poisoning kills around 150,000 people each year [6]. It is not clear how many of the cases are due to OP insecticides, but they probably account for about 2/3 of deaths. The references need to be updated.

Reply: We thank the reviewer for correcting this statement, and providing the numbers. As reviewer suggested, we have modified statements and included new references in the revised manuscript.

Comment R1-11: “Pesticides inhibit AChE, which leads to severe neuronal/cognitive dysfunction, breathing disorders, loss of endurance, and death when exposed to high acute doses.” Only organophosphorus and carbamate insecticides inhibit AChE, not all pesticides.

Reply: We thank the reviewer for correcting this statement. As reviewer suggested we have modified the statement as “Organophosphates and carbamate insecticides inhibit AChE, which leads to severe neuronal/cognitive dysfunction, breathing disorders, loss of endurance, and death when exposed to high acute doses.

Reviewer # 2. We thank the reviewer for finding the results of our work is interesting and considered for publishing in Nature Communications. The reviewer raised one critical point that has been addressed.

Comment R2-1: The chemistry used in this study is simple, by silanization on the hydroxyl groups of cellulose. However, the author did not show proof of the reaction of oxime onto cellulose. FTIR or solid state NMR could be useful to prove the covalent linkage of the oxime with cellulose.

Reply: We thank the reviewer for this critical suggestion. As reviewer suggested, we have conducted solid state NMR experiments to provide the proof for covalent linkage of the oxime with cellulose.

To provide further evidence for covalent functionalization of silyl-oxime on the fabric, solid state Cross Polarization Magic Angle Spinning (CP/MAS) ^{13}C -NMR studies were performed. Pyridinium ring carbons in silyl-oxime compound give peaks between 141 – 148 δ ppm (**Supplementary Figure S2**). Therefore, to find the presence of silyl-oxime upon covalent functionalization on the fabric, ^{13}C -NMR was recorded for Normal fabric, Oxime-fabric and Oxime-fabric after extensive washes to remove non-covalent bound silyl-oxime. Spectra in **Figure R2 and R3 (Supplementary Figure S3 and S4)** show the presence of silyl-oxime on the fabric. The peaks corresponding to pyridinium group are completely absent in spectra of normal fabric, whereas these peaks were found in spectra of oxime fabric. Subsequently, oxime fabric was extensively washed with both detergent and methanol in which silyl-oxime is soluble. However, despite of extensive washing did not remove pyridinium peaks suggesting that silyl-oxime has been covalently attached to the fabric.

Additionally, presence of silyl-oxime was quantified using bromophenol (BPB) assay (Figure 2g). Quantification of pyridinium of oxime fabric through BPB assay has shown that even after 50 cycles of washes did not remove silyl-oxime from fabric (Figure 2g). Therefore cumulatively, ^{13}C -NMR and BPB assay data suggesting that silyl-oxime covalently conjugated to the fabric.

Figure R2. Solid state Cross Polarization Magic Angle Spinning (CP/MAS) ^{13}C -NMR spectra. a, spectra of Normal Fabric, **b,** Oxime Fabric after curing, and **c,** Oxime Fabric after extensive washing. Peaks corresponding to pyridinium carbons (blue arrow) were present only on Oxime-fabric and they remain present after washing suggesting the presence of silyl-oxime on Oxime Fabric.

We have included the current data in the revised supplementary information (**Supplementary Fig. S3 and S4**), and modified the text accordingly.

Reviewer # 3. We thank the reviewer for critically reading our manuscript and providing their constructive feedback and appreciating saying a well-written article. Please see below for our response to each point that has been raised by the referee.

Comment R3-1: ABSTRACT: Lines 23-26: “Oxime-fabric, when stitched as a bodysuit and facemask, upon contact efficiently deactivates pesticides (organophosphates, carbamates, and pyrethrins), and prevents exposure through skin and nasal routes, respectively”. How can oxime-fabric catalytically deactivate pyrethrins? Please, clarify it.

Reply: We thank the reviewer for discussing this point. Pyrethrins are carboxylate-based insecticides. Akin to oxime mediated hydrolysis of phosphate esters in organophosphates, N-O⁻ nucleophile of oxime can hydrolyze carboxylate esters, therefore, catalytically deactivate pyrethrins. For example; cypermethrin is one of the most common insecticides used in agricultural formulations, schematic of oxime mediated cypermethrin hydrolysis has been shown in **Figure R4**.

We have carried our HPLC analysis to provide the evidence that Oxime-fabric can deactivate pyrethrins. Same concentration of Cypermethrin was added on Normal fabric and Oxime fabric, upon incubation for 5 mins, cypermethrin was extracted from fabric and analyzed using HPLC. HPLC chromatograms (**Figure R5**) suggests that incubation on normal fabric did not affect cypermethrin, whereas upon incubation on Oxime fabric, cypermethrin has been degraded. The quantification results (**Figure R6**) suggest that we could extract the cyperthrin quantitatively from normal fabric, whereas, >95% of cypermethrin that was incubated on oxime fabric was degraded. Cumulatively, these results suggest that Oxime fabric has the ability to deactivate pyrithrins. These data has been included in the revised supplementary information (**Supplementary Fig. S7, S8 and S9**), and the text has been modified accordingly.

Figure R5. Oxime mediated hydrolysis of cypermethrin. HPLC chromatograms of standard cypermethrin and samples extracted after incubating on either Normal Fabric or Oxime Fabric. (Green arrows represents peaks of cypermethrin).

Red line: Cypermethrin standard (250 µg/mL).

Black line: Extracted sample after cypermethrin (250 µg/mL) incubated on Normal Fabric.

Blue line: Extracted sample after cypermethrin (250 µg/mL) incubated on Oxime Fabric.

Comment R3-2: INTRODUCTION: Lines 42-45: In the 1st paragraph: “Therefore, inhibition of AChE due to pesticide exposure causes neurological dysfunction, breathing disorder, paralysis, and death in severe cases from the Asian-Pacific region are repeatedly exposed to pesticides during farming”, the authors use pesticide as insecticide synonym. Please, correct it.

Reply: We thank the reviewer pointing out this mistake. As suggested, we have corrected it, not only in this place, we carefully checked throughout the manuscript and corrected wherever is appropriate.

Comment R3-3: Lines 88-102: The introduction should address the specific research problem, the gaps, specify objectives and what is intended to be done. However, the last paragraph of the introduction describes and discusses the results and presents conclusions. Please, move the last paragraph to suitable sections (results and discussion).

Reply: We thank the reviewer for this suggestion. We do agree that we had detailed results in this paragraph. In the revised manuscript, we have deleted 12 lines and shortened the paragraph to describe the overall outcome. The detailed results were moved to appropriate sections.

Comment R3-4: Lines 43-49: In “Therefore, inhibition of AChE due to pesticide exposure causes neurological dysfunction, breathing disorder, paralysis, and death in severe cases... Electroencephalograms of OP-exposed humans and monkeys revealed that in addition to their acute toxicity effects, OPs have also been reported to have long-term effects. Additionally, agricultural workers have reported neuro-behavioural changes even two years after a single exposure to OP”, the author included an old reference on the sarin chemical weapon toxic effect in SNC. However, there are recent literature on organophosphate insecticide long-term exposure and neurological disorders. See in: <https://doi.org/10.1016/j.tox.2022.153407>

Reply: We thank the reviewer for suggesting the new reference. We have included this reference in the revised manuscript.

Comment R3-5: Lines 51-54: About “Typically used pesticides/insecticides include organophosphates (methyl parathion, ethyl paraoxon, methamidophos, oxydemeton-methyl, chlorpyrifos, malathion, and primifos-methyl), carbamates (carbaryl, aldicarb, carbofuran, and ferbam), and pyrethrins/ pyrethroids”. Please, remove pesticide in the sentence.

Reply: We apologize for the mistake. We have removed it.

Comment R3-6: “Chemical pesticides are primarily categorized as organophosphate esters, carbamates, and pyrethrins.” Chemical pesticide classes include types of insecticides, fungicides, herbicides so on. Please, in the sentence above change “pesticide” to “insecticides”.

Reply: As suggested by the reviewer, we have changed it to insecticides.

Comment R3-7: Lines 113-118: About “Chemical pesticides are primarily categorized as organophosphate esters, carbamates, and pyrethrins. Among them, organophosphates constitute >70% of the total pesticides used in the Indian agricultural fields. Thus, developing a robust nucleophile that can hydrolyse organophosphate esters, carbamates, and pyrethrin esters will enable the deactivation of pesticides. A wide range of α -nucleophiles are known to hydrolyze organophosphates.”: The results section summarizes and presents the study findings to put them in context with research question. The study's data should be presented in a logical sequence without bias or interpretation. Please, consider removing them and including them in the discussion section.

Reply: We thank the reviewer for this suggestion. As the reviewer suggested, we have modified the text in the revised manuscript.

Comment R3-8: Still on lines 113-114: "...organophosphates constitute >70% of the total pesticides used in the Indian agricultural fields.". Please, include the reference.

Reply: We the reviewer suggested, we have included the references in the revised manuscript.

Comment R3-9: Lines 147-149: About: “To test the robustness of Oxime-fabric to deactivate a wide range of pesticides, a similar assay was performed using carbamates and a wide range of commercial organophosphate formulations that contain organophosphates and pyrethrins as well.”: In the supplementary material, I did not find any results on the catalysis of insecticide formulation containing pyrethroids by oxime fabric.

Reply: We thank the reviewer for pointing out this mistake. We have purchased the commercial formulations that containing pyrethrins. The following formulations contain pyrethrins. **Raise-505** (Chlorpyrifos 50% + Cypermethrin 5% EC, Agastya Agro Ltd., Muraharipally, Telangana, India), and **Profex Super** (Profenofos 40% + Cypermethrin 4% EC, Nagarjuna Agrichem Ltd., Punjagutta, Hyderabad, India). We have conducted ex vivo experiments using Franz diffusion assays to demonstrate that Oxime fabric can prevent

commercial formulations that contain pyrethrins. The data is given in revised **Supplementary Fig. S6**, and for the convenience it is given below Figure.

Figure: Oxime-fabric deactivates commercial pesticide formulations, and prevents AChE inhibition, *ex vivo*. **a**, Photograph image of commercial formulations used in this study. **b**, The efficacy of Oxime-fabric to prevent commercial insecticide formulations-induced AChE inhibition, an *ex vivo* assay was performed using rat blood. AChE containing rat blood was placed in the acceptor chamber. In the donor chamber, various insecticides carbamate (**c**), MPT (**d**), Macacid-50 (**e**), Aalphos (**f**), Raise-505 (**g**), and Profex Super (**h**) were added in the presence of normal fabric or Hazmat suit fabric or Oxime-fabric. Active AChE was measured in unexposed native blood and 3 hr post addition of pesticide formulations in these groups. The normal fabric and Hazmat suit fabric could not prevent diffusion of commercial pesticide formulations into the acceptor chamber, which resulted in significant inhibition of

AChE activity. On the contrary, Oxime-fabric could hydrolyze carbamate and insecticides in commercial formulations before they diffuse, hence prevented the insecticide-induced inhibition of AChE. Data are mean \pm s.d. ($n = 4$, from independent experiments). For **c-h**, P values were determined by ordinary one-way ANOVA with Tukey's post hoc analysis by GraphPad PRISM 9, and exact P values are indicated. ns = not significant. Source data are provided as a Source Data file.

Comment R3-10: Lines 126-127: About: "Due to the presence of the pyridinium group, the BPB assay gave the quantitative oxime presence on the fabric (Fig. 2a, b)". The results in figure 2b show the presence or not of color in the fabric. Consider only including Figure 2a as indicative of quantitative results.

Reply: We thank the reviewer for pointing this out. In fact, in Figure 2a, b describe only schematic and photographic images of dye-bound fabric. The quantitative data of oxime on fabric is presented in **Figure 2c**. Therefore, to reflect figure reference given as **Fig. 2a-c** in the revised manuscript.

Comment R3-11: Lines 181-192: The discussion of Figure 2 f and h is presented elsewhere after the presentation of complementary Figure 5. Please, include the results in the correct order.

Reply: As the reviewer suggested, we have moved results presented in Figure 2 f and h to right after Figure 2 e. Now, the results are in the correct order.

Comment R3-12: Lines 147-162: The results description of supplementary tables 3 and 4 is very wordy. I think the author should be direct.

Reply: We thank the reviewer for this suggestion. As suggested, we have revised the writeup. We have merged previous Supplementary Fig. S3 and S4 into one figure to give the clarity. Accordingly, we have revised the text in the revised manuscript. Now, it is direct and readers can follow the results easily.

Comment R3-13: Lines: 218-221: The results section needs to be straightforward. In “Pesticides disrupt the functioning of the cholinergic nervous system by irreversibly inhibiting AChE, which hydrolyses acetylcholine at the synapse. Hence, inhibition of AChE leads to overstimulation of cholinergic receptors, resulting in neuronal excitotoxicity, dysfunction, and disrupted signaling at the neuromuscular junctions.”: Please. Consider removing the sentence from the results section and including it in the discussion.

Reply: We agree with the reviewer that results section needs to be straightforward. However, in this particular place, we would like to retain the sentence. Because this sentence gives the context and importance of the experiment that was carried out. It would be easier for readers to follow the results presented in this section after reading the first sentence. Therefore, we have retained the sentence in the revised manuscript.

Comment R3-14: Figure 4 b is not mentioned in the results section (photos of paws).

Reply: We thank the reviewer for suggesting this point. In the revised manuscript, we have mentioned about photos of paws (**Fig. 4b**).

Comment R3-15: Lines 270-273: About “Oxime-fabric prevents repeated pesticide exposure-induced mortality in vivo. As farmers in the field repeatedly get exposed to multiple doses of pesticides, which could be lethal and can cause mortality, we investigated the robustness of Oxime-fabric to prevent, ethyl paraoxon (EPx)-induced mortality in rats”. Please include references that affirm occupational exposure to multiple pesticide doses could be lethal and can cause mortality.

Reply: As the reviewer suggested we have provided references in the revised manuscript. Additionally, we also modified the statement as following “As farmers in the field repeatedly get exposed to multiple doses of insecticides, which could cause long-term adverse health effects”.

Comment R3-16: Lines 240-242: About “Pesticide-induced AChE inhibition causes the accumulation of the neurotransmitter acetylcholine at the synapse, leading to disturbed signaling at neuromuscular junctions and overstimulation of muscles”. The author uses pesticide as a synonym for insecticide. Please, replace pesticide with insecticide.

Reply: As the reviewer suggested, it has been changed.

Comment R3-17: In tropical developing countries, people often place their clothes in the sun to dry. The authors did not evaluate the oxime fabric's thermal stability and photodegradation. The authors should discuss these issues in the limitations section.

Reply: We thank the reviewer for pointing this critical point. In fact, we have evaluated the thermal stability and stability under the sun light of Oxime fabric has been evaluated. During the curing process itself Oxime fabric get exposed to 120 °C, and still retains its activity.

Furthermore, to test the stability of Oxime fabric under the sun light, Oxime fabric was exposed to sun light for three days, and subsequently investigated its efficacy to deactivate insecticide, MPT. The data in **Figure R7** suggest that even after exposed to the sun light Oxime fabric retained its activity and prevented MPT-induced AChE inhibition. The data is included in the revised supplementary information (**Supplementary Fig. S5**).

Figure R7. Oxime-fabric retains its activity after exposed to sunlight for three days, *ex vivo*. The stability of Oxime-fabric upon exposure to sunlight has been tested. The efficacy of Oxime-fabric to prevent insecticide-induced AChE inhibition, before and after exposed to sunlight for three days remained the same, which suggests that Oxime-fabric could be used under sunlight as farmers do. Data are mean \pm s.d. ($n = 4$, from independent experiments). P values were determined by ordinary one-way ANOVA with Tukey's post hoc analysis by GraphPad PRISM 9, and exact P values are indicated. ns = not significant. Source data are provided as a Source Data file.

Comment R3-18: It is worth mentioning some diseases associated with long-term insecticide exposure in the discussion section.

Reply: As the reviewer suggested, we have mentioned some diseases associated with long-term insecticide exposure in the discussion section along with references.

Comment R3-19: *P*-nitrophenol is the catalytic product of organophosphates. This substance is absorbed through the skin (www.epa.gov/sites/default/files/2016-09/documents/4-nitrophenol.pdf; <https://www.atsdr.cdc.gov/toxprofiles/tp50.pdf>). Would *p*-nitrophenol be toxic to rats and humans? It is important to discuss this issue as a possible study limitation or future issue to investigate.

Reply: We thank the reviewer for this critical point. However, due to the adequate documentation of toxic effects of *p*-nitrophenol (pNP), we have not studied pNP toxicity. According to a report by Agency for Toxic Substances and Disease Registry, U.S. Public Health Service, no lethality in rats when exposed to 4,033 mg/m³ of atmospheric concentration (inhalation route) of pNP for 4 hrs or when exposed to 2,119 mg/m³ for 6 hours/day for 10 days. The oral LD₅₀ of pNP in rats was reported to be 620 mg/kg when administered with corn oil. When rabbits were exposed dermally to 5,000 mg/kg of pNP as saline suspension, no lethality was observed. Additionally, no deaths were observed when rats were treated dermally to pNP at concentrations as high as 250 mg/kg/day for 120 days. None of these doses match possible pNP exposure to even after lethal doses of EPx; thus, pNP produced during decontamination of EPx will have a significantly lower health hazard.

Reference: Vernot, E. H., MacEwen, J. D., Haun, C. C., & Kinkead, E. R. Acute toxicity and skin corrosion data for some organic and inorganic compounds and aqueous solutions. *Toxicology and Applied Pharmacology* 1977;42(2):417–423.

<https://www.atsdr.cdc.gov/toxprofiles/tp50.pdf>

Comment R3-20: Lines 393-395: Please include references to “Due to frequent spraying, farmers repeatedly get exposed to pesticides, which causes chronic toxicity and mortality”.

Reply: As the reviewer suggested we have included the references in the revised manuscript.

Once again, we thank all reviewers for their time for critically reading the manuscript and providing their valuable suggestions, which have significantly strengthened the manuscript.

REVIEWER COMMENTS

Reviewer #1 (Remarks to the Author):

Thank you for addressing my concerns.

1.

It is a relief that you have actually looked at formulated pesticides, including OP/pyrethroid combination products

2.

You have replied to my first point about the novel use of oximes but have not (that I can see) revised the manuscript.

Please consider add a second section to this sentence:

“Earlier, we demonstrated that pyridine-2-aldoxime can hydrolytically deactivate a wide range of organophosphates, carbamates, and pyrethrins,²³ in addition to their better known effect of reactivating OP-inhibited acetylcholinesterase (Eyer 2003)”

3.

You may wish to consider using the more commonly used name for the oxime of pralidoxime (versus the much less widely used name ‘pyridine-2-aldoxime’)

4.

Regarding the response to my comment that AChE inhibition is not irreversible because oximes reverse inhibition.

This point still stands. Ageing does occur but takes hours, so for the first 12 hrs following dimethyl OP exposure and ~130 hrs following diethyl OP exposure, inhibited AChE can be reactivated if sufficient oximes are given.

Reviewer #2 (Remarks to the Author):

The authors have addressed all comments and I would recommend publication.

Reviewer #3 (Remarks to the Author):

Dear Authors,

Thank you for the responses sent. However, I would like to request more information on issues that I consider critical:

1. Particularly, the aspect of pyrethroids hydrolysis by AChE raises significant concerns. It is imperative for the authors to substantiate their hypothesis with relevant literature. The sole reference provided by the author in support of this hypothesis lacks conclusive evidence. (<https://www.ncbi.nlm.nih.gov/pmc/articles/PMC6192682/>). The illustration of pyrethroids hydrolysis by AChE does not sufficient. It is essential for the author to reference pertinent literature that specifically investigates the interaction of AChE with pyrethroids. Please, clarify it.

2. Another complex issue arises from the results concerning the pyrethroids hydrolysis by oxime tissue. The authors failed to test formulations containing a singular pyrethroid active ingredient, opting instead for formulations with a high concentration of organophosphates and a minimal amount of pyrethroids. Please clarify it.

Other minor questions:

1. It is imperative that authors review all manuscript references: For example, in the sentence "India is one of the largest countries in the world that heavily uses insecticides, and farmers from the Asian-Pacific region are repeatedly exposed to insecticides during farming^{8,9}. ": the reference Narayan, S. et al. Genetic variability in ABCB1, occupational pesticide exposure, and Parkinson's disease. *Environ. Res.* 143, 98–106 (2015). is not adequate. Other references are very old. I suggest including more current references.

2. The authors included references on the health effects of organophosphates used as chemical weapons. There are differences in toxicity between chemical weapons and pesticides for agricultural use. I suggest that the authors include references on the health effects of agricultural organophosphates.

3. Regarding the sentence "In India, organophosphate based insecticides are 76% of the total pesticides, as compared to 44% globally". Please include the reference.

We thank all the reviewers for their positive response, critical feedback and recommending for publication. As few minor points were raised, we have answered those queries and revised the manuscript accordingly. Please see below for our response to each point that the reviewer has raised.

Reviewer # 1. We thank the reviewer for recommending for the publication. Please see below for our response to each point that has been raised by the referee.

Comment R1-1: It is a relief that you have actually looked at formulated pesticides, including OP/pyrethroid combination products

Reply: We are glad to know that reviewer is satisfied with our experimental approach.

Comment R1-2: You have replied to my first point about the novel use of oximes but have not (that I can see) revised the manuscript. Please consider add a second section to this sentence:

“Earlier, we demonstrated that pyridine-2-aldoxime can hydrolytically deactivate a wide range of organophosphates, carbamates, and pyrethrins,²³ in addition to their better known effect of reactivating OP-inhibited acetylcholinesterase (Eyer 2003)”

Reply: As reviewer suggested, we have modified the sentence and added the reference.

Comment R1-3: You may wish to consider using the more commonly used name for the oxime of pralidoxime (versus the much less widely used name ‘pyridine-2-aldoxime’)

Reply: We thank the reviewer for this suggestion. Accordingly, we have replaced ‘pyridine-2-aldoxime’ with ‘pralidoxime’ in the throughout manuscript.

Comment R1-4: Regarding the response to my comment that AChE inhibition is not irreversible because oximes reverse inhibition.

This point still stands. Ageing does occur but takes hours, so for the first 12 hrs following dimethyl OP exposure and ~130 hrs following diethyl OP exposure, inhibited AChE can be reactivated if sufficient oximes are given.

Reply: We thank and agree with the reviewer for this point, therefore, we have deleted the word “irreversibly”, and changed the sentence to “Pesticides disrupt the functioning of the cholinergic nervous system by inhibiting AChE, which hydrolyses acetylcholine at the synapse.”

Reviewer # 2. We glad to know that reviewer is satisfied with the revised manuscript, and we thank the reviewer for recommending for publication.

Reviewer # 3. We thank the reviewer for critically reading our manuscript and providing constructive feedback. Please see below for our response to each point that has been raised by the referee.

Comment R3-1: Particularly, the aspect of pyrethroids hydrolysis by AChE raises significant concerns. It is imperative for the authors to substantiate their hypothesis with relevant literature. The sole reference provided by the author in support of this hypothesis lacks conclusive evidence. (<https://www.ncbi.nlm.nih.gov/pmc/articles/PMC6192682/>). The illustration of pyrethroids hydrolysis by AChE does not sufficient. It is essential for the author to reference pertinent literature that specifically investigates the interaction of AChE with pyrethroids. Please, clarify it.

Reply: We thank the reviewer for raising this point. Sharma et al (*International Journal of Nutrition, Pharmacology, Neurological Diseases* 4, 104-111 (2014)) have reported that cypermethrin-induced neurotoxicity through inhibition of AChE activity. In this article, they have proposed a model for interaction between cypermethrin and AChE (shown below). In this model, the authors have proposed that pi-pi interactions between Phe and Trp present in active site of AChE and diphenyl-ether of cypermethrin are feasible. Such interaction maybe further stabilized by hydrogen bonding of carbonyl of CN of cypermethrin. Subsequently, cypermethrin may bind with Ser, possibly through transesterification (figure adopted from the reference).

Figure 1: Schematic representation of possible binding sites of acetylcholinesterase enzyme with cypermethrin

Note: This figure was taken from the original publication, and is provided only for review, not to include in the manuscript.

Additionally, they have reported that pyrethrins such as cypermethrin can inhibit AChE activity in rat brains. To unambiguously demonstrate cypermethrin-mediated inhibition of AChE activity, AChE containing rat brain homogenate was exposed to cypermethrin and AChE activity was quantified using Ellman's assay. As a positive control, we compared with MPT-induced AChE inhibition. Data in **Figure R1 (Supplementary Fig. S10)** clearly suggests that cypermethrin can inhibit AChE activity significantly akin to MPT. However, data indicates that cypermethrin is less efficient AChE inhibitor than organophosphate based MPT.

Figure R1. Cypermethrin-mediated AChE inhibition. The efficacy of MPT and cypermethrin to inhibit AChE activity has been quantified with an *ex vivo* assay using rat brain homogenate. MPT (2.5 μM) and Cypermethrin (75 μM) were significantly inhibited AChE activity. *P* values were determined by ordinary one-way ANOVA with Tukey's post hoc analysis by GraphPad PRISM 9, and exact *P* values are indicated.

Comment R3-2: Another complex issue arises from the results concerning the pyrethroids hydrolysis by oxime tissue. The authors failed to test formulations containing a singular pyrethroid active ingredient, opting instead for formulations with a high concentration of organophosphates and a minimal amount of pyrethroids. Please clarify it.

Reply: As reviewer suggested, now we have conducted *ex vivo* Franz diffusion experiments with a singular pyrethroid active ingredient, cypermethrin, and demonstrated that upon exposure, cypermethrin significantly inhibited AChE. Subsequently, our data suggests that Normal fabric could not prevent cypermethrin-mediated AChE inhibition, whereas Oxime-fabric is efficient in deactivating cypermethrin and prevents cypermethrin-mediated AChE inhibition (**Figure R2** and **Supplementary Fig. S11**).

Minor Comments:

Comment R3-m1: It is imperative that authors review all manuscript references: For example, in the sentence "India is one of the largest countries in the world that heavily uses insecticides, and farmers from the Asian-Pacific region are repeatedly exposed to insecticides during farming^{8,9}. ": the reference Narayan, S. et al. Genetic variability in ABCB1, occupational pesticide exposure, and Parkinson's disease. *Environ. Res.* 143, 98–106 (2015). is not adequate. Other references are very old. I suggest including more current references.

Reply: We thank the reviewer for this suggestion. We have reviewed all references, and included updated with recent references. Below references are included.

Jørs, E., Neupane, D. & London, L. Pesticide poisonings in low- and middle-income countries. *Environ Health Insights.* **12**, 1-3 (2018)

Damalas, C. A. & Koutroubas, S. D. Farmers' exposure to pesticides: Toxicity types and ways of prevention. *Toxics* **4**, 1 (2016)

Mancini, F., Van Bruggen, A. H., Jiggins, J. L., Ambatipudi, A. C. & Murphy, H. Acute pesticide poisoning among female and male cotton growers in India. *Int J Occup Environ Health.* **11**, 221–232 (2005)

Tomenson, J. A. & Matthews, G. A. Causes and types of health effects during the use of crop protection chemicals: data from a survey of over 6,300 smallholder applicators in 24 different countries. *Int Arch Occup Environ Health.* **82**, 935–949 (2009)

Comment R3-m2: The authors included references on the health effects of organophosphates used as chemical weapons. There are differences in toxicity between chemical weapons and pesticides for agricultural use. I suggest that the authors include references on the health effects of agricultural organophosphates.

Reply: As referee suggested, we have included the references on the health effects of agricultural organophosphates.

Campos, E., dos Santos Pinto da Silva, V., Sarpa Campos de Mello, M. & Barros Otero, U. Exposure to pesticides and mental disorders in a rural population of Southern Brazil. *Neurotoxicology* **56**, 7-16 (2016)

Damalas, C. A. & Koutroubas, S. D. Farmers' exposure to pesticides: Toxicity types and ways of prevention. *Toxics* **4**, 1 (2016)

Comment R3-m3: Regarding the sentence "In India, organophosphate based insecticides are 76% of the total pesticides, as compared to 44% globally". Please include the reference.

Reply: We have included the reference in the revised manuscript.

Aktar, Md. W., Sengupta, D. & Chowdhury, A. Impact of pesticides use in agriculture: their benefits and hazards. *Interdiscip Toxicol.* **2**, 1-12 (2009).

Once again, we thank all reviewers for their time for critically reading the manuscript and providing their valuable suggestions, which have significantly strengthened the manuscript.

Reviewers' comments:

Reviewer #1 (Remarks to the Author):

This is a very large piece of interesting work. However the terminology is inaccurate or unhelpful at times. There is much confusion in the use of 'pesticides', 'insecticides' and 'OP insecticides'.

Only OP and carbamate insecticides inhibit AChE, pyrethroids do not inhibit AChE. Therefore the assays shown in Fig Suppl 6 panels G and H are only addressing the roles of the OP insecticides chlorpyrifos and profenofos. The assay does not indicate whether the oxime cloth has hydrolysed the pyrethroids. This is suggested by your figures R5 and R6 in the response to reviewers (now Supplementary Fig. S7, S8 and S9) but not shown by this experiment. This should be clarified.

It will be helpful if you could go through the paper and revise the terms in places to be more precise:

1. Cypermethrin is not a permethrin but a pyrethroid. Permethrins are used rarely as insecticides. Pyrethroids are much more widely used in both agriculture and public health. In this paper, please refer only 'pyrethroids' as this is accurate.

2. When you are referring to AChE inhibition, please do not use the term 'pesticides'. Instead, where you are looking at methyl parathion or ethyl parathion/oxon, refer to 'OP insecticides'. More generally, you can refer to OP and carbamate insecticides inhibiting AChE.

Your paper focuses on OP inhibition of AChE; carbamates (carbaryl) and pyrethroids are relatively peripheral to your work.

Regarding the Abstract, you have made the latter point clear ["Organophosphates and carbamate insecticides inhibit AChE, which leads to ..."] but not the latter ["efficiently deactivates pesticides (organophosphates, carbamates, and pyrethrins)] – this should be pyrethroids.

It would also be helpful at first mention of methyl parathion to say that it is an OP insecticide: Perhaps: "We report that, in the presence of Oxime-fabric, AChE inhibition induced by the OP insecticide methyl parathion (MPT) in blood and tissues is reduced significantly."

Minor – specific examples causing confusion

1.
"while unintentional poisoning kills around 150,000 people each year1 -"

This should be 'intentional self-poisoning' not 'unintentional'

2.
"Organophosphates (OPs) inhibit the enzyme AChE," should be "Organophosphate (OP) and carbamate insecticides inhibit the enzyme AChE,"

3.
"These results suggest that Oxime-fabric could reduce pesticide-induced AChE inhibition by chemically deactivating insecticides"
Should be

"These results suggest that Oxime-fabric could reduce OP or carbamate insecticide-induced AChE inhibition by chemically deactivating insecticides"

4.
"Pesticides disrupt the functioning of the cholinergic nervous system by irreversibly inhibiting AChE, which hydrolyses acetylcholine at the synapse."

Should be

""OP and carbamate insecticides disrupt the functioning of the cholinergic nervous system by irreversibly inhibiting AChE, which hydrolyses acetylcholine at the synapse.""

5.

"Insecticide-induced AChE inhibition causes the accumulation of the neurotransmitter"

Should be:

"OP insecticide-induced AChE inhibition causes the accumulation of the neurotransmitter"

6.

"we investigated the robustness of Oxime-fabric to prevent insecticide, ethyl paraoxon (EPx)-induced mortality in rats."

Should be:

"we investigated the robustness of Oxime-fabric to prevent ethyl paraoxon (EPx, an activated OP insecticide)-induced mortality in rats."

7.

"Typically used insecticides include organophosphates (methyl parathion, ethyl paraoxon,"

Ethyl paraoxon is not an insecticide but an activated metabolite of the OP insecticide parathion (or ethyl parathion). Please delete from this list.

Reviewer #3 (Remarks to the Author):

Dear Authors,

While I appreciate your effort in justify the action of pyrethroids on acetylcholinesterase, I must express concern regarding the lack of consolidated literature supporting your hypothesis. The action of insecticides has been widely studied over the last 40 years, and I am not aware of literature that suggests such a mechanism of action for pyrethroids. I had highlighted imperative to include specialized literature (manuscripts) indicating the inhibition of acetylcholinesterase by pyrethroids and incorporate these references into the paper. Relying solely on a single experimental paper may not provide adequate substantiation for your claims. Furthermore, I must emphasize that a theoretical model of molecular interaction alone is insufficient to substantiate your findings. If these questions raised were not addressed with substantial literature supporting their findings, it is very worrying to assert something that is not supported in the literature and for that reason alone I consider that the article should not be published.

We thank both reviewers for their critical feedback. As few minor points were raised, we have answered those queries and revised the manuscript accordingly. Please see below for our response to each point that the reviewer has raised.

Reviewer # 1. We thank the reviewer for appreciating the work. Please see below for our response to each point that has been raised by the referee.

Comment R1-1: This is a very large piece of interesting work. However the terminology is inaccurate or unhelpful at times. There is much confusion in the use of ‘pesticides’, ‘insecticides’ and ‘OP insecticides’.

Reply: We apologize for the confusion in the use of terminology. We went through the manuscript, and we have extensively corrected to represent right terminology for organophosphate insecticides. For clarity, we have completely removed ‘pesticide’ word, and modified in the title as well.

Comment R1-2: Only OP and carbamate insecticides inhibit AChE, pyrethroids do not inhibit AChE. Therefore the assays shown in Fig Suppl 6 panels G and H are only addressing the roles of the OP insecticides chlorpyrifos and profenofos. The assay does not indicate whether the oxime cloth has hydrolysed the pyrethroids. This is suggested by your figures R5 and R6 in the response to reviewers (now Supplementary Fig. S7, S8 and S9) but not shown by this experiment. This should be clarified.

Reply: In this work, we primarily focused on OP and carbamate insecticides induced AChE inhibition. To avoid potential confusion, we now have removed the entire data related to pyrethroids (Supplementary Fig. S7-S11 have been removed). Accordingly, in the revised manuscript text/discussion also we have confined to OP and carbamate induced AChE inhibition, and removed pyrethroid part.

Comment R1-3: It will be helpful if you could go through the paper and revise the terms in places to be more precise:

1. Cypermethrin is not a permethrin but a pyrethroid. Permethrins are used rarely as insecticides. Pyrethroids are much more widely used in both agriculture and public health. In this paper, please refer only ‘pyrethroids’ as this is accurate.

Reply: As suggested, we have changed the terminology, and corrected accordingly. However, we have removed pyrethroids part from the manuscript.

Comment R1-4: 2. When you are referring to AChE inhibition, please do not use the term ‘pesticides’. Instead, where you are looking at methyl parathion or ethyl parathion/oxon, refer to ‘OP insecticides’. More generally, you can refer to OP and carbamate insecticides inhibiting AChE.

Reply: We thank the reviewer for this clarification. As suggested, we have changed this terminology throughout the manuscript.

Comment R1-5: Your paper focuses on OP inhibition of AChE; carbamates (carbaryl) and pyrethroids are relatively peripheral to your work.

Reply: We do agree with the referee opinion, and in the revised version, we have removed pyrethroids part completely. The revised version has focused on OP and carbamate insecticides induced AChE inhibition.

Comment R1-6: Regarding the Abstract, you have made the latter point clear [“Organophosphates and carbamate insecticides inhibit AChE, which leads to ...”] but not the latter [“efficiently deactivates pesticides (organophosphates, carbamates, and pyrethrins)] – this should be pyrethroids.

Reply: We have made changes as suggested.

Comment R1-7: It would also be helpful at first mention of methyl parathion to say that it is an OP insecticide:

Perhaps: “We report that, in the presence of Oxime-fabric, AChE inhibition induced by the OP insecticide methyl parathion (MPT) in blood and tissues is reduced significantly.”

Reply: We have made changes as suggested.

Minor points:

Comment R1-8: “while unintentional poisoning kills around 150,000 people each year¹” This should be ‘intentional self-poisoning’ not ‘unintentional’

Reply: Thank you, typo has been corrected.

Comment R1-9: “Organophosphates (OPs) inhibit the enzyme AChE,” should be “Organophosphate (OP) and carbamate insecticides inhibit the enzyme AChE,”

Reply: As suggested, it has been corrected.

Comment R1-10: “These results suggest that Oxime-fabric could reduce pesticide-induced AChE inhibition by chemically deactivating insecticides”

Should be:

“These results suggest that Oxime-fabric could reduce OP or carbamate insecticide-induced AChE inhibition by chemically deactivating insecticides”

Reply: As suggested, it has been corrected.

Comment R1-11: “Pesticides disrupt the functioning of the cholinergic nervous system by irreversibly inhibiting AChE, which hydrolyses acetylcholine at the synapse.”

Should be:

“OP and carbamate insecticides disrupt the functioning of the cholinergic nervous system by irreversibly inhibiting AChE, which hydrolyses acetylcholine at the synapse.”

Reply: As suggested, it has been corrected.

Comment R1-12: “Insecticide-induced AChE inhibition causes the accumulation of the neurotransmitter”

Should be:

“OP insecticide-induced AChE inhibition causes the accumulation of the neurotransmitter”

Reply: As suggested, it has been corrected.

Comment R1-13: “we investigated the robustness of Oxime-fabric to prevent insecticide, ethyl paraoxon (EPx)-induced mortality in rats.”

Should be:

“we investigated the robustness of Oxime-fabric to prevent ethyl paraoxon (EPx, an activated OP insecticide)-induced mortality in rats.”

Reply: As suggested, it has been corrected.

Comment R1-14: “Typically used insecticides include organophosphates (methyl parathion, ethyl paraoxon,” Ethyl paraoxon is not an insecticide but an activated metabolite of the OP insecticide parathion (or ethyl parathion). Please delete from this list.

Reply: As suggested, it has been corrected.

Additionally, based on the feedback, we made changes in the title to reflect correct terminology

Previous title:

Oxime-functionalized anti-pesticide fabric reduces pesticide exposure through dermal and nasal routes, and prevents pesticide-induced neuromuscular-dysfunction and mortality

Revised title:

Oxime-functionalized anti-insecticide fabric reduces insecticide exposure through dermal and nasal routes, and prevents insecticide-induced neuromuscular-dysfunction and mortality

Reviewer # 3. We thank the reviewer for critical suggestion.

Comment R3-1:

Dear Authors,

While I appreciate your effort in justify the action of pyrethroids on acetylcholinesterase, I must express concern regarding the lack of consolidated literature supporting your hypothesis. The action of insecticides has been widely studied over the last 40 years, and I am not aware of literature that suggests such a mechanism of action for pyrethroids. I had highlighted imperative to include specialized literature (manuscripts) indicating the inhibition of acetylcholinesterase by pyrethroids and incorporate these references into the paper. Relying solely on a single experimental paper may not provide adequate substantiation for your claims. Furthermore, I must emphasize that a theoretical model of molecular interaction alone is insufficient to substantiate your findings. If these questions raised were not addressed with substantial literature supporting their findings, it is very worrying to assert something that is not supported in the literature and for that reason alone I consider that the article should not be published.

Reply: We thank the reviewer for raising this critical point. We agree with the reviewer. In this manuscript, we are primarily focusing on organophosphate-insecticide induced AChE inhibition, and prevention of organophosphate insecticides using oxime-fabric. Pyrethroids part was peripheral.

Therefore, we are removing the portions of pyrethroids related data/discussion. Supplementary Fig. S7-S11 have been removed. In this manuscript, we are confined to organophosphate and carbamate insecticides induced AChE inhibition. Accordingly, we have revised the manuscript, and supplementary data. The revised manuscript will completely eliminate the confusion about pyrethroids effect on AChE. This will be systematically investigated in the future work.

These changes completely eliminate the ambiguity in the current work.

Once again, we thank both reviewers for their time for critically reading the manuscript and providing their valuable suggestions, which have significantly strengthened the manuscript.